# Improved analyses of GWAS summary statistics by reducing data heterogeneity and errors

Wenhan Chen[1,2], Yang Wu [1], Zhili Zheng[1], Ting Qi[1,3,4], Peter M. Visscher [1], Zhihong Zhu[1] & Jian Yang [1,3,4✉]

Summary statistics from genome-wide association studies (GWAS) have facilitated the development of various summary data-based methods, which typically require a reference sample for linkage disequilibrium (LD) estimation. Analyses using these methods may be biased by errors in GWAS summary data or LD reference or heterogeneity between GWAS and LD reference. Here we propose a quality control method, DENTIST, that leverages LD among genetic variants to detect and eliminate errors in GWAS or LD reference and heterogeneity between the two. Through simulations, we demonstrate that DENTIST substantially reduces false-positive rate in detecting secondary signals in the summary-data-based conditional and joint association analysis, especially for imputed rare variants (false-positive rate reduced from >28% to <2% in the presence of heterogeneity between GWAS and LD reference). We further show that DENTIST can improve other summary-data-based analyses such as fine-mapping analysis.

[1] Institute for Molecular Bioscience, The University of Queensland, Brisbane, QLD 4072, Australia. [2] Epigenetics Research Laboratory, Genomics and Epigenetics Theme, Garvan Institute of Medical Research, Sydney, NSW 2010, Australia. [3] School of Life Sciences, Westlake University, Hangzhou, Zhejiang 310024, China. [4] Westlake Laboratory of Life Sciences and Biomedicine, Hangzhou, Zhejiang 310024, China. ✉email: jian.yang@westlake.edu.cn

Genome-wide association studies (GWASs) have been extraordinarily successful in uncovering genetic variants associated with complex human traits and diseases[1,2]. Summary statistics available from GWASs have facilitated the development of various summary-data-based methods[3] such as those for fine-mapping[4–11], imputing summary statistics at untyped variants[12,13], estimating SNP-based heritability[14–16], assessing causal or genetic relationship between traits[17–19], prioritizing candidate causal genes for a trait[20–23], and polygenetic risk prediction[8,24,25]. Most of the summary-data-based methods require linkage disequilibrium (LD) structure of the variants used, which are unavailable in the summary data but can be estimated from a reference cohort with individual-level genotypes, assuming a homogeneous LD structure between the GWAS and reference cohorts. Hence, summary-data-based analyses can be biased by not only errors in the GWAS and LD reference data sets but also differences between them for the following reasons. First, there are often errors in GWAS summary statistics resulting from data generation and analysis processes (e.g., genotyping/imputation errors and genetic variants with mis-specified effect alleles)[26,27], some of which are not easy to detect, even if individual-level data are available. Second, there is often heterogeneity between data sets (e.g., between the discovery GWAS and LD reference) because of differences in ancestry, and genotyping platform, imputation quality, and analysis pipeline. Although the recommended practice is to use an ancestry-matched reference cohort, samples with similar ancestries, such as populations of European ancestry, can still have discernable differences in LD structure[28], and the effects of such differences on summary-data-based analyses are largely unexplored.

In this study, we propose a quality control (QC) method to identify errors in GWAS summary data and heterogeneity between summary data and LD reference by testing the difference between the observed z-score of each variant and its predicted value from the surrounding variants. The method has been implemented in a software tool named DENTIST (detecting errors in analyses of summary statistics). We show by simulation that DENTIST can effectively detect errors of several kinds after standard QC. We then demonstrate the utility of DENTIST as an additional QC step for multiple, frequently-used, summary data-based methods, including the conditional and joint analysis (COJO)[6] of summary statistics, FINEMAP[9], and LD score regression[14].

## Results

**Overview of the DENTIST method.** Details of the methodology can be found in the Methods section. In brief, we first use a sliding window approach to divide the variants into 2 Mb segments with a 500 kb overlap between two adjacent segments. Within each segment, we randomly partition variants into two subsets, S1 and S2, with an equal number of variants, and apply the statistic below to test the difference between the observed z-score of a variant $i$ ($z_i$) in S1 and its predicted value ($\widetilde{z}_i$) based on z-scores of an array of variants $\mathbf{t}$ in S2 (Methods).

$$T_{d(i)} = \frac{(z_i - \widetilde{z}_i)^2}{1 - \mathbf{R}_{it}\mathbf{R}_{tt}^{-1}\mathbf{R}_{it}'} \text{ with } \widetilde{z}_i = \mathbf{R}_{it}\mathbf{R}_{tt}^{-1}\mathbf{z_t} \quad (1)$$

where $\mathbf{z_t}$ is a vector of z-scores of variants $\mathbf{t}$ in S2, and $\mathbf{R}$ is the LD correlation matrix calculated from a reference sample with $\mathbf{R}_{tt}$ being the LD between variants $\mathbf{t}$ and $\mathbf{R}_{it}$ being the LD between variant $i$ and variants $\mathbf{t}$. Note that methods that leverage LD to predict GWAS test statistic of a variant (i.e., $\widetilde{z}_i$) from test statistics of its adjacent variants (i.e., $\mathbf{z_t}$) have been developed in prior work[12,13]. $T_d$ follows approximately a $\chi^2$ distribution with 1 degree of freedom. We denote $P$ value calculated from $T_d$ by $P_{\text{DENTIST}}$. A significant difference between the observed and predicted z-scores indicates error(s) in the discovery GWAS or LD reference, and/or heterogeneity between them. If the difference between $z_i$ and $\widetilde{z}_i$ is due to error in $z_i$, the power of $T_d$ depends on how $z_i$ deviates from its true value and how well variant $i$ is tagged by variants $\mathbf{t}$. Following prior work[15], we use the truncated singular value decomposition (SVD) method to compute a pseudo inverse of $\mathbf{R}_{tt}$ to improve computational stability (Methods).

One challenge for the DENTIST method is that errors can be present in both S1 and S2, and errors in S2 can inflate $T_d$ statistics of the variants in S1. To mitigate this issue, we propose an iterative variant partitioning approach (Methods). In each iteration, we partition the variants into two subsets (S1 and S2) at random, remove variants with $P_{\text{DENTIST}} < 5 \times 10^{-8}$ from the variant group with $P_{\text{GWAS}} > 0.01$, and apply the same proportion of variants removed to the variant group with $P_{\text{GWAS}} < 0.01$ (note: the proportion of variants removed is capped at 0.5% in each group). This step is to create a more reliable set of variants for the next iteration. The problematic variants are prioritized and filtered out in the first few iterations so that the prediction of $\widetilde{z}_i$ (Eq. 1) becomes more accurate in the following iterations. We set

---

**Table 1 A summary of the GWAS data sets with individual-level genotypes used in the present study.**

| Data set identifier | Sample size | Genotyping platform | Imputation reference | HWE $P$ threshold | MAC/MAF threshold | # Variants (MAF > 0.01) | # Variants (MAF > 0.001) |
|---|---|---|---|---|---|---|---|
| HRS | 8,557 | Illumina Omni 2.5 | 1KGP | $10^{-6}$ | 5 (MAC) | 11,326,629 | 15,608,163 |
| ARIC | 7,703 | Affymetrix 6.0 | 1KGP | $10^{-6}$ | 5 (MAC) | 8,627,204 | 13,356,572 |
| UK10K-WGS | 3,642 | WGS ( > 30x coverage) | NA | $10^{-6}$ | 3 (MAC) | 8,325,293 | 12,798,760 |
| UK10K-1KGP | 3,642 | Illumina Core Exome | 1KGP | $10^{-6}$ | 3 (MAC) | 9,351,188 | 14,398,774 |
| 1KGP-EUR | 503 | WGS (~7.4x coverage) | NA | $10^{-6}$ | 0.01 (MAF) | 9,417,956 | 9,417,956 |
| HRS-3K | 3,000 | Illumina Omni 2.5 | 1KGP | $10^{-6}$ | 0.01 (MAF) | 45,901 | 45,901 |
| UKBv3 | 348,577 | Affymetrix Axiom | HRC + UK10K | $10^{-6}$ | 0.0001 (MAF) | 8,569,717 | 13,072,891 |
| UKBv3-329K | 328,577 | Affymetrix Axiom | HRC + UK10K | $10^{-6}$ | 0.0001 (MAF) | 8,569,717 | 13,072,891 |
| UKBv3-8K | 8,000 | Affymetrix Axiom | HRC + UK10K | $10^{-6}$ | 0.0001 (MAF) | 9,351,188 | 14,398,774 |
| UKBv3-20K | 20,000 | Affymetrix Axiom | HRC + UK10K | $10^{-6}$ | 0.0001 (MAF) | 9,351,188 | 14,398,774 |
| UKB-8K-1KGP | 8,000 | Affymetrix Axiom | 1KGP | $10^{-6}$ | 0.0001 (MAF) | 7,349,687 | 9,972,916 |

*1KGP* 1000 Genome Project Phase III data, *WGS* whole genome sequencing, *HRC* Haplotype Reference Consortium, *HM3* HapMap 3 Project.

the number of iterations to 10 in practice. All variants with $P_{DENTIST} < 5 \times 10^{-8}$ are removed in the final step without the constraints.

**Detecting simulated errors in GWAS data after standard QC.** To assess the performance of DENTIST in detecting errors, we simulated GWAS data with genotyping errors and allelic errors (i.e., variants with the effect allele mis-labeled) using whole genome sequence (WGS) data on chromosome 22 of 3,642 unrelated individuals from the UK10K project after QC[29,30] (denoted by UK10K-WGS). A descriptive summary of the data sets used in this study can be found in Table 1 and the Methods section. We performed standard QC on the imputed data to exclude variants with an imputation INFO score <0.3 or a Hardy-Weinberg Equilibrium (HWE) P value $< 10^{-6}$. We simulated a trait affected by 50 common, causal variants with effects drawn from $N(0, 1)$, which together explained 20% of the phenotypic variation (proportion of variance explained by a causal variant, denoted by $q^2$, was 0.4%, on average). Prior to the simulations with errors, we performed a benchmark simulation (i.e., simulating a scenario without errors and applying DENTIST using the discovery GWAS sample as the LD reference) to show that the DENTIST test statistics were well calibrated, given the small fraction of variants removed by DENTIST in each simulation replicate and the small proportion of causal variants removed across simulation replicates, in the absence of errors and LD heterogeneity (Supplementary Fig. 1).

We then simulated genotyping and allelic errors at 0.5% randomly selected variants respectively. Genotyping errors at each of these variants were simulated by altering the genotypes of a certain proportion ($f_{error} = 0.05, 0.1$ or $0.15$) of randomly selected individuals, and allelic error of each of the variants was introduced by swapping the effect allele by the other allele. We acknowledge that genotyping errors are unlikely to be a major concern for summary data-based analyses. The purpose of adding genotyping errors to the simulation was to quantify the ability of DENTIST to detect errors in general and to explore the possibility of using DENTIST as an additional QC step for GWAS analyses with individual data to detect genotyping errors not detected by the standard GWAS QC pipeline. The simulation was repeated 200 times with the causal and erroneous variants re-sampled in each simulation. We then ran DENTIST using UK10K-WGS or an independent sample (UKB-8K-1KGP) as the LD reference after standard QC of the discovery GWAS: removing variants with an HWE P value $< 10^{-6}$ using the individual-level data or $\Delta AF > 0.1$ where $\Delta AF$ is the difference in allele frequency (AF) between the summary data and reference sample. The independent sample UKB-8K-1KGP refers to a set of 8,000 unrelated individuals from the UK Biobank[31] (UKB) with variants imputed from the 1000 Genomes Project (1KGP). The statistical power was measured by the proportion of erroneous variants in the data that can be detected by DENTIST. We also computed the fold enrichment in probability of an erroneous variant being detected by DENTIST compared to a random guess (i.e., the ratio of the percentage of true erroneous variants in the variants detected by DENTIST to that in all variants).

When using UKB-8K-1KGP as the reference, ~45% of the genotyping and ~95% of the allelic errors could be removed by the standard GWAS QC (Supplementary Table 1). After the standard QC, DENTIST was able to detect ~27% of the remaining genotyping and ~73% of the remaining allelic errors (Fig. 1 and Supplementary Table 2), with only ~0.2% variants being removed in total (Supplementary Table 3). The fold enrichment of errors in the removed variants was 366 for allelic errors and of 137 for genotyping errors (Supplementary Table 4). Notably, the power

to detect allelic errors was ~73% for variants with MAF > 0.45, compensating the low power of the $\Delta AF$ approach in this MAF range (~16%). When restricted to variants passing the genome-wide significance level (i.e., $P_{GWAS} < 5 \times 10^{-8}$), the power of DENTIST increased to ~93% for the genotyping errors and 100% for the allelic errors (Supplementary Table 2). The power also varied with the genotyping error rate ($f_{error}$), e.g., the power in the $f_{error} = 0.05$ scenario was, on average, lower than that for $f_{error} = 0.15$ (Fig. 1c and Supplementary Table 2). When using UK10K-WGS as the reference (mimicking the application of DENTIST in a scenario where individual-level data of the discovery GWAS are available), the power remained similar, but the fold enrichment was much higher compared to that using UKB-8K-1KGP (Supplementary Tables 4 and 5). The percentage of causal variants removed from DENTIST was 0.19% using in-sample LD from UK10K-WGS, < 1% using out-of-sample LD from UKB-8K-1KGP (Supplementary Table 3). Moreover, using the same simulation setting, we explored the choice of the parameter $\theta_k$ (i.e., the proportion of eigenvectors retained in the truncated SVD method; see Methods for details) and reference sample size ($n_{ref}$); the results suggested that a choice of $\theta_k = 0.5$ and $n_{ref} \geq 5000$ is appropriate in practice (Supplementary Fig. 2). In addition, the power of DENTIST to detect genotyping and allelic errors was robust to the randomness of variant splitting (Supplementary Fig. 3). Altogether, these results demonstrate the power of DENTIST to identify allelic and genotyping errors after the standard QC, suggesting that DENTIST can complement existing QC filters for either individual- or summary-level GWAS data. Nevertheless, we acknowledge that this simulation did not cover the full complexity of real case scenarios, which may involve multiple independent samples with heterogeneous LD structures caused by several factors, such as imputation errors or ancestry mismatches (Supplementary Fig. 4). These cases are difficult to mimic in this simulation but will be assessed in the following analyses.

**Applying DENTIST to COJO with simulated phenotypes.** COJO is a method that uses GWAS summary statistics and reference LD to run a conditional and joint multi-SNP regression analysis[6]. We used simulations to assess the performance of COJO, as implemented in GCTA v1.92.3 (ref. [32]), in the presence of heterogeneity between the discovery GWAS and LD reference samples before and after DENTIST filtering. To mimic the reality that causal signals are often not perfectly captured by imputed variants, we simulated a phenotype affected by one or two sequence variants using WGS data (i.e., UK10K-WGS) and performed association analyses using imputed data of the same individuals (imputing 312,264 variants, in common with those on an SNP array, to the 1KGP[30,33]; denoted by UK10K-1KG). More specifically, we first randomly selected one or two sequence variants with MAF ≥ 0.01 (denoted by common-causal) or $0.01 > MAF \geq 0.001$ (denoted by rare-causal) from a locus as causal variant(s) to generate a phenotype (note: MAF > 0.001 is equivalent to minor allele count > 7 in this sample). The proportion of phenotypic variance explained by the causal variant (denoted by $q^2$) was set to 2% to achieve similar power to a scenario with $q^2 = 0.03\%$ and $n = 250,000$ (note: the mean $q^2$ of the 697 height-associated variants discovered in Wood et al.[34] is 0.03%) because the power of GWAS is determined by $nq^2/(1 - q^2)$. Then, we ran a GWAS using UK10K-1KGP and performed COJO analyses using multiple LD references, including the discovery GWAS sample (i.e., UKB-8K-1KGP), the Health Retirement Study (HRS)[35], and the Atherosclerosis Risk In Communities (ARIC) study[36], with different degrees of ancestral differences with UK10K-1KGP (Supplementary Fig. 5). For a fair

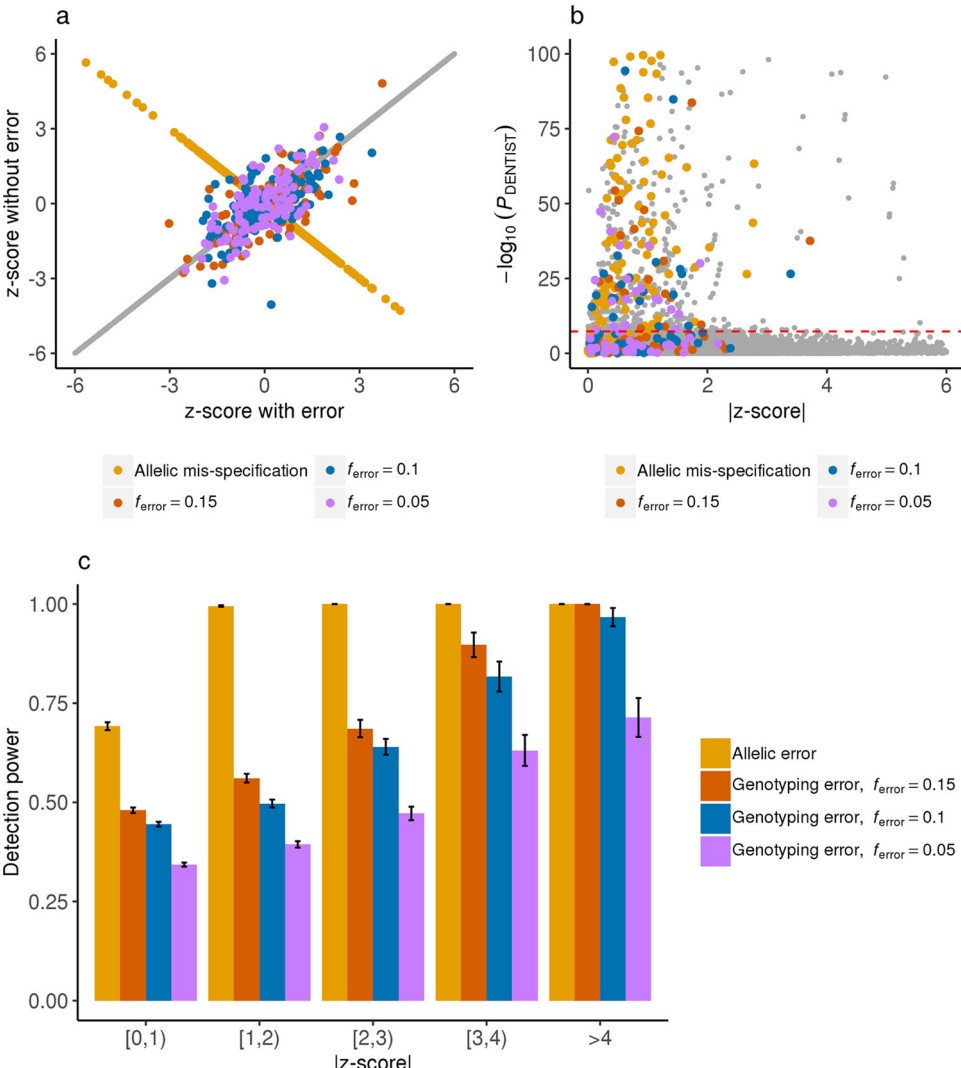

**Fig. 1 Detecting simulated allelic and genotyping errors using DENTIST.** We assessed the power of DENTIST in detecting allelic and genotyping errors by simulation. There are three levels of genotyping error rate ($f_{error}$=0.15, 0.1, or 0.05), defined as the proportion of individuals with erroneous genotypes for a variant. **a** is a plot of the GWAS z-scores from data with simulated errors against those from data without such errors. The gray dots in the diagonal represents z-scores of variants without errors. In (**b**), the DENTIST P values are plotted against the absolute values of the GWAS z-scores for all the variants. The horizontal dashed line corresponds to $P = 5 \times 10^{-8}$. **c**, the height of each bar represents the power in a |z-score| bin estimated from 200 simulation replicates, with the error bar representing ± one s.e. of the estimate. Source data are provided as a Source Data file.

comparison, only the variants shared between these reference samples were included. We repeated both the common-causal and rare-causal simulations 100 times for each autosome and computed the false-positive rate (FPR, i.e., the frequency of observing two COJO signals in the scenario where there was only one causal variant) and power (the frequency of observing two COJO signals in the scenario where there were two distinct causal variants with LD $r^2 < 0.1$ between them). It should be noted that the false-positive COJO signals defined here are not false associations but falsely claimed as jointly associated (also known as quasi-independent) signals.

When using the discovery GWAS sample as the LD reference, the FPR of COJO was 0.1% for common-causal and 0.2% for rare-causal (Fig. 2 and Table 2), which can be regarded as a benchmark for comparison as there was no data heterogeneity in this case. The FPRs were higher than the expected values because the causal variants were not perfectly tagged by the imputed variants (Supplementary Table 6). When using UKB-8K-1KGP (i.e., 1KGP-imputed data of 8,000 UKB participants with similar

ancestry to the UK10K participants as shown in Supplementary Fig. 5) as the LD reference, the FPR was close to the benchmark for common-causal (1%) and slightly inflated for rare-causal (2.7%) (Fig. 2). After DENTIST filtering (using UKB-8K-1KGP as the LD reference), the FPR for rare-causal decreased to 1.6%. When using LD computed from European-American individuals in HRS or ARIC, the FPR of COJO was strongly inflated in the whole MAF range: >7% for common-causal and >28% for rare-causal, likely because of the difference in ancestry between HRS/ARIC and UK10K-1KGP. DENTIST could effectively control the FPR of COJO to <1% for common-causal and <3.5% for rare-causal (Fig. 2 and Table 2). We performed additional simulations with three causal variants at a locus and observed similar patterns of changes in FPR before and after DENTIST QC (Supplementary Fig. 6). Taken together, the FPR of COJO was reasonably well controlled for common variants but substantially inflated for rare variants, especially when there was a difference in ancestry between the GWAS and LD reference samples, and most of the false-positive COJO signals could be removed by DENTIST.

The power of COJO (without DENTIST) using in-sample LD from UK10K-1KGP or out-of-sample LD from the other references were similar: 77–80% for common-causal and 26–30% for rare-causal (Table 2). The low power for rare causal variants was because they were poorly tagged by imputation (Supplementary Table 6). DENTIST filtering caused a <1% loss of power for common-causal and 3–4% for rare-causal (Table 2), which is acceptable because the reduction in FPR was larger than that in power, especially for rare variants.

**Applying DENTIST to COJO for real phenotypes.** Having assessed the performance of DENTIST in COJO analyses by simulation, we then applied it to COJO analyses for height in the UKB. The height GWAS summary statistics ($n = 328,577$) were generated from a GWAS analysis of all the unrelated individuals of European ancestry (denoted by UKBv3-329K) except 20,000 individuals (denoted by UKBv3-20K), deliberately left-out to be used as a non-overlapping LD reference. Genotype imputation of the UKB data was performed by the UKB team with most of the variants imputed from the Haplotype Reference Consortium

(HRC)[37]. We performed COJO analyses with a host of references: overlapping in-sample references with sample sizes ($n_{ref}$) varying from 10,000 to 150,000, non-overlapping in-sample references including UKBv3-8K ($n = 8,000$) and UKBv3-20K (containing UKBv3-8k), and out-of-sample references including ARIC and HRS (Table 1). We excluded from the analysis variants with MAF < 0.001 to ensure sufficient minor alleles for rare variants in reference samples with $n_{ref} < 10k$. We first performed a COJO analysis using the actual GWAS sample as the reference and identified 1,279 signals from variants with MAFs >0.01, and 1,310 signals from variants with MAF > 0.001 (Table 3). These results can be regarded as a benchmark. When using the overlapping in-sample LD references, the number of COJO signals first decreased as $n_{ref}$ increased and then started to stabilize when $n_{ref}$ exceeded 30,000 (Supplementary Fig. 7). The results from using the two non-overlapping in-sample references (UKBv3-8K and UKBv3-20K) were comparable to those from using the overlapping in-sample references with similar sample sizes (Table 3 and Supplementary Table 7) because the non-overlapping in-sample references, despite being excluded from the GWAS, were consistent with the GWAS sample with respect to ancestry, data collection, and analysis procedures.

When using LD from an out-of-sample reference (either HRS or ARIC), there was substantial inflation in the number of COJO signals compared to the benchmark (by 15.5–16.1% for common variants and 18.7–25.6% for all variants) (Table 3), with a few variants in weak LD with those identified from the benchmark analysis (Supplementary Fig. 8). The results from using the two out-of-sample references became more consistent with the benchmark after DENTIST filtering, with the inflation reduced to 5.0–7.2% for common variants and 3.9–4.4% for all variants, comparable to the results using an in-sample LD reference with a similar sample size (Table 3). Polygenic score analysis shows that the reduction in the number of COJO signals owing to DENTIST QC had almost no effect on the accuracy of using the COJO signals to predict height in HRS (Supplementary Table 8), suggesting the redundancy of the signals removed by DENTIST. We further found that compared to using the imputed data from ARIC or HRS, using UK10K-WGS ($n = 3,642$) as the reference showed lower inflation (10.9% for common variants and <8.5% for all variants) before DENTIST QC but larger inflation after DENTIST QC (Table 3), suggesting a large reference sample size is essential even for WGS data. In all the DENTIST analyses above, the total number of removed variants varied from 0.03% to 1.4% (Supplementary Table 9). All these results are consistent with what we observed from simulations, demonstrating the effectiveness of DENTIST in eliminating heterogeneity between GWAS and LD reference samples.

We further applied DENTIST to Educational Attainment (EA), Coronary Artery Disease (CAD), Type 2 Diabetes(T2D), Crohn's Disease (CD), Major Depressive Disorder (MDD), Schizophrenia

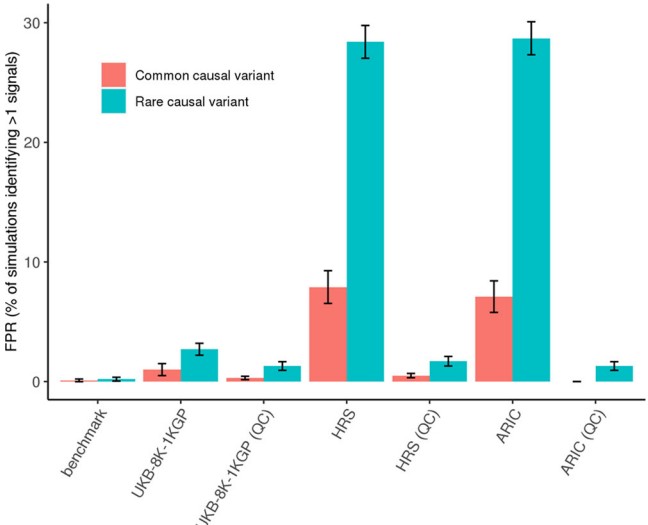

**Fig. 2 FPR of COJO with and without DENTIST.** Based on simulations with one causal variant, we assessed the FPR of COJO when performed with and without DENTIST-based QC (FPR is defined as the frequency of observing more than one COJO signal in the scenario where only one causal variant was simulated). The x-axis labels indicate the LD reference samples, and those performed after DENTIST QC are labeled with "QC" in the parentheses. The height of each bar shows the FPR estimated from 2200 simulation replicates, with each error bar representing ± one s.e. of the estimate. Source data are provided as a Source Data file.

| Table 2 FPR and power of COJO before and after DENTIST-based QC in simulations. | | | | |
|---|---|---|---|---|
| Analysis method (LD reference) | FPR for common-causal (%) | FPR for rare-causal (%) | Power for common-causal (%) | Power for rare-causal (%) |
| Benchmark | 0.1 ± 0.11 | 0.2 ± 0.16 | 78.8 ± 0.9 | 30.6 ± 1.4 |
| COJO without DENTIST (UKB-8K-1KGP) | 1.0 ± 0.50 | 2.7 ± 0.50 | 79.0 ± 0.9 | 30.6 ± 3.0 |
| COJO with DENTIST (UKB-8K-1KGP) | 0.2 ± 0.13 | 1.63 ± 0.43 | 80.0 ± 1.2 | 29.4 ± 1.6 |
| COJO without DENTIST (HRS) | 7.9 ± 1.37 | 28.4 ± 1.37 | 78.5 ± 1.2 | 27.6 ± 2.9 |
| COJO with DENTIST (HRS) | 0.5 ± 0.18 | 2.68 ± 0.55 | 78.8 ± 1.2 | 23.0 ± 1.6 |
| COJO without DENTIST (ARIC) | 7.1 ± 1.32 | 28.7 ± 1.38 | 77.6 ± 0.9 | 26.3 ± 2.9 |
| COJO with DENTIST (ARIC) | 0.4 ± 0.17 | 3.50 ± 0.63 | 76.8 ± 1.0 | 23.2 ± 1.6 |
| Benchmark: COJO analysis using the discovery GWAS as the reference without DENTIST. Shown are mean ± standard error. | | | | |

**Table 3 Numbers of COJO signals from analyses of the UKB height summary data using different LD reference samples with and without DENTIST-based QC.**

| LD reference (sample size) | No DENTIST & MAF > 0.01 | DENTIST & MAF > 0.01 | No DENTIST & MAF > 0.001 | DENTIST & MAF > 0.001 |
|---|---|---|---|---|
| Benchmark | 1279 | N/A | 1310 | N/A |
| UKBv3-20K ($n = 20,000$) | 1296 (1.3%) | 1288 (0.7%) | 1313 (0.2%) | 1324 (1.0%) |
| UKBv3-8K ($n = 8000$) | 1337 (4.5%) | 1326 (3.6%) | 1337 (2.0%) | 1325 (1.1%) |
| HRS ($n = 8557$) | 1477 (15.5%) | 1344 (5.0%) | 1555 (18.7%) | 1362 (3.9%) |
| ARIC ($n = 7703$) | 1485 (16.1%) | 1372 (7.2%) | 1645 (25.6%) | 1368 (4.4%) |
| UK10K-WGS ($n = 3,642$) | 1417 (10.8%) | 1419 (10.9%) | 1473 (12.4%) | 1422 (8.5%) |

Benchmark: COJO analysis using the discovery GWAS (UKBv3-329K) as the reference without DENTIST. The inflation rate relative to to the benchmark is shown in the parentheses.

**Table 4 A summary of the 10 published GWAS summary data sets used in this study.**

| Trait (abbreviation) | $n$ | $n_{cases}$ | $n_{controls}$ | Year | PUBMED ID |
|---|---|---|---|---|---|
| Educational Attainment (EA) | 766,345 | / | / | 2018 | 30038396 |
| Coronary Artery Disease (CAD) | 547,261 | 122,733 | 424,528 | 2017 | 29212778 |
| Type 2 Diabetes(T2D) | 898,130 | 74,124 | 824,006 | 2018 | 30297969 |
| Crohn's Disease (CD) | 20,883 | 5956 | 14,927 | 2015 | 26192919 |
| Major Depressive Disorder (MDD) | 480,359 | 135,458 | 344,901 | 2018 | 29700475 |
| Schizophrenia (SCZ) | 35,802 | 11,260 | 24,542 | 2018 | 29483656 |
| Ovarian Cancer (OC) | 66,450 | 25,509 | 40,941 | 2018 | 28346442 |
| Breast Cancer (BC) | 228,951 | 12,2977 | 105,974 | 2018 | 29059683 |
| Height | ~700,000 | / | / | 2018 | 30124842 |
| Body Mass Index (BMI) | ~700,000 | / | / | 2018 | 30124842 |

$n$: sample size; $n_{cases}$ and $n_{controls}$: numbers of cases and controls, respectively.

(SCZ), Ovarian Cancer (OC), Breast Cancer (BC), Height, and Body Mass Index (BMI) using GWAS summary data from prior studies[38–46] (Table 4) and three LD reference samples (i.e., ARIC, HRS, and UKBv3-8K). Since these published studies focus on common variants (rare variants are unavailable in most of the data sets), we used an MAF threshold of 0.01 in this analysis. When using ARIC as the LD reference, the proportion of variants removed by DENTIST QC ranged from 0.07% (BMI) to 0.86% (CAD) with a median of 0.21% (Supplementary Table 10), and the reduction in the number of COJO signals for common variants ranged from 0% (OC and MDD) to 12.5% (CAD) with a median of 1.6% (Supplementary Table 11). The results from using the other two references were similar (Supplementary Tables 10 and 11).

**Improved FINEMAP analysis by DENTIST.** FINEMAP[9] is a method used for prioritizing causal variants from GWAS summary statistics. We assessed the effect of DENTIST QC on an FINEMAP analysis using the data simulated above with one, two or three common causal variants at a 1-Mb locus. The impact of the use of an out-of-sample LD reference on FINEMAP v1.4 was evaluated by the estimated number of causal variants before or after DENTIST QC, in comparison with the benchmark analysis using in-sample LD. We observed that FINEMAP using out-of-sample LD tended to overestimate the number of causal variants (Supplementary Figs. 9 and 10). For example, given a posterior inclusion probability threshold of 95%, FINEMAP with out-of-sample LD from the HRS cohort provided the correct number of causal variants in 78.4%, 49.8%, and 44.8% of the simulation replicates in the one-, two-, and three-causal scenarios, respectively (Fig. 3a). After DENTIST QC, the proportions of times when FINEMAP identified the correct number of causal variants increased substantially to 87.3%, 78.5%, and 63.4% in the one-, two-, and three-causal scenarios, respectively, almost comparable to the corresponding proportions (i.e., 91.4%, 87.2%, and 68.2%) from the benchmark analysis (Fig. 3a). The results remained

similar when using ARIC as the LD reference (Fig. 3b). These results demonstrate a substantial improvement of the performance of FINEMAP after DENTIST QC.

**LD score regression analysis with DENTIST.** LD score regression (LDSC) is an approach which was originally developed to distinguish polygenicity from population stratification in GWAS summary data set by a weighted regression of GWAS $\chi^2$ statistics against LD scores computed from a reference[14] but has often been used to estimate the SNP-based heritability ($h^2_{SNP}$). We investigated the impact of DENTIST on LDSC v1.0.0 using the height GWAS summary data generated using the UKBv3-329K sample along with several reference samples including four imputation-based samples (i.e., the discovery GWAS sample, HRS, ARIC and UKB-8K-1KGP) and two WGS-based samples, i.e., UK10K-WGS and the European individuals from the 1KGP (1KGP-EUR). We performed the one- and two-step LDSC analyses using LD scores of the variants, in common with those in the HapMap3, computed from each of the references before and after DENTIST-based QC. Note that DENTIST was performed for all common variants but only those overlapped with HapMap3 were included in the LDSC analyses.

Using the discovery GWAS sample as the reference, the estimates of $h^2_{SNP}$ and regression intercept from the one-step LDSC were 46% (SE = 0.02) and 1.13 (SE = 0.04) respectively (Supplementary Table 12). When using the other reference samples, the results were very close to the benchmark except for a noticeably larger estimate of regression intercept using HRS (1.24, SE = 0.04). After DENTIST filtering, the intercept estimate using HRS decreased to 1.18 (SE = 0.04) with a small difference in $\hat{h}^2_{SNP}$ (increased from 45 to 47%) (Supplementary Table 12). To better understand the effect of DENTIST QC on LDSC in this case, we plotted the mean $\chi^2$-statistic against the mean LD score across the LD score bins. We found that the GWAS mean $\chi^2$-statistic in the bin with the smallest mean LD score deviated from the value

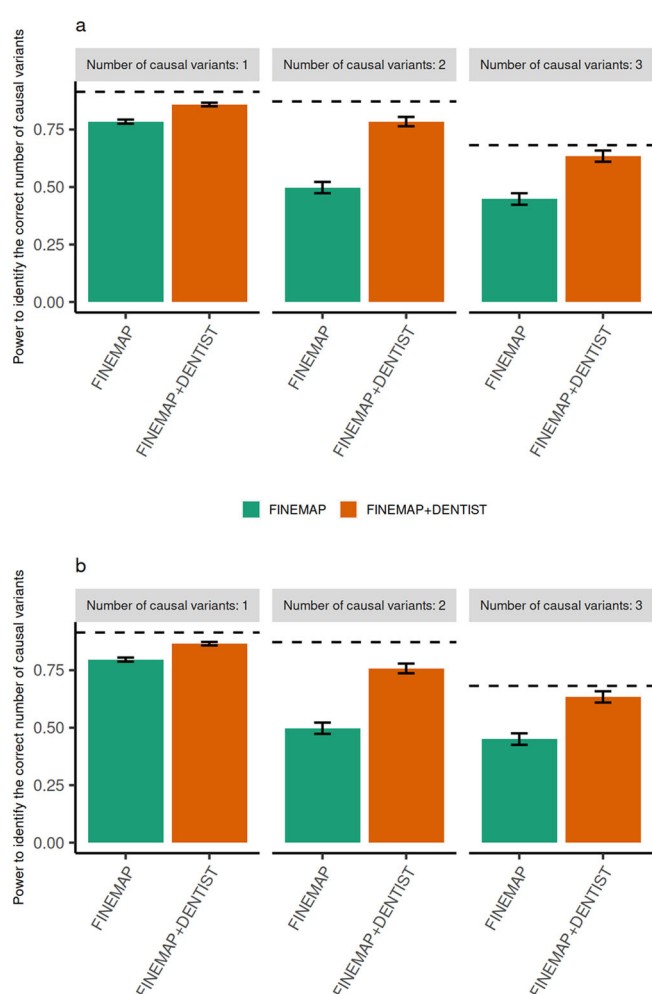

**Fig. 3 Power of FINEMAP with and without DENTIST QC.** The power of FINEMAP is measured by the percentage of simulation replicates in which the number of causal variants is correctly identified under the simulation scenario with 1, 2, or 3 causal variants. In **a**, LD reference = HRS. In **b**, LD reference = ARIC. The horizontal dashed line in each bar plot shows the benchmark, i.e., the power of FINEMAP when using in-sample LD. The height of each bar plot represents the power estimated from the pooled results of 2200 replicates. The error bar represents s.e. estimated from binomial distribution. Source data are provided as a Source Data file.

expected from a linear relationship between the LD score and $\chi^2$-statistic (Supplementary Fig. 11), and the deviation was removed by filtering out a small proportion of variants with very small LD scores but large $\chi^2$-statistic by DENTIST (Supplementary Fig. 12). These results show how the quality of LD reference can impact the LDSC analysis, which is typically unknown a priori and varied across different reference samples. We also re-ran the analyses using the two-step LDSC, where the intercept was estimated using the variants with $\chi^2$ values < 30 in the first step and constrained in the second step to estimate $h^2_{SNP}$ using all the variants. Compared to the one-step approach, the two-step approach provides larger estimates of the intercepts and smaller estimates of $h^2_{SNP}$ either before or after DENTIST filtering. In addition, we applied LDSC to the 10 published GWAS data sets mentioned above (Table 4) using ARIC, UKB-8K-1KG and UK10K-WGS as the reference. The improvement of LDSC by DENTIST QC was small particularly for traits whose LDSC intercepts were close to 1

before DENTIST QC (Supplementary Table 13), demonstrating the robustness of LDSC to data heterogeneity and errors.

## Discussion

In this study, we developed DENTIST, a QC tool for summary data-based analyses, which leverages LD from a reference sample to detect and filter out problematic variants by testing the difference between the observed z-score of a variant and a predicted z-score from the neighboring variants. From simulations and real data analyses, we show that some of the commonly used analyses can be biased to various extents in the presence of data heterogeneity, e.g., inflated number of COJO signals or overestimated number of causal variants from FINEMAP. For most of these analyses, DENTIST-based QC can mitigate the biases to some extent. DENTIST has the unique feature to detect heterogeneity between a GWAS summary data set and an LD reference. The benefit of using DENTIST as a QC tool has been demonstrated in the three case studies above, but we believe that it can potentially be applied to all GWAS summary data-based analyses that require a LD reference such as other fine-mapping methods[7,47], and joint modeling of all variants for polygenic risk prediction[24,25,48]. In addition to dealing with data heterogeneity, the power of DEN-TIST also comes from detecting genotyping or imputation errors that passed standard QCs, making it a complementary method to existing QC methods. There could be another source of hetero-geneity if the GWAS analysis is adjusted for covariates such as age, sex, and genetic PCs, but the genotype data in the reference sample are not covariate-adjusted. However, such a contribution is likely to be negligible given the near perfect correlation between LD correlation values computed from the UK10K genotype data before and after covariate adjustment (Supplementary Fig. 13).

Our results suggest that summary-data-based analyses are gen-erally well calibrated in the absence of data heterogeneity but biased otherwise despite the use of multiple standard QCs. For example, heterogeneity between the discovery GWAS and LD reference caused inflated FPR of COJO (Table 3). Also, we found that the FPR of COJO for rare variants was much higher than that for common variants likely because rare variants are more difficult to impute such that they are more likely to have discrepancy in LD between two imputed data sets. It should be clarified that the false-positive COJO signals as defined here are not false associations but falsely claimed as jointly significant associations. DENTIST sub-stantially reduced false-positive detections from COJO analyses, especially for rare variants, even when there was a difference in ancestry between the GWAS and LD reference samples. This extends the utility of COJO, which was originally developed for common variants, to rare variants. This message is important for the field because more and more GWASs and meta-analyses have started to include imputed/sequenced rare variants. It is also encouraging to see the substantial improvement of FINEMAP after DENTIST QC, implying the utility of DENTIST for fine-mapping methods. Among all the methods tested, LDSC was the least affected by errors or data heterogeneity (Supplementary Fig. 11).

The improvement of a summary data-based analysis before and after DENTIST QC depends on how susceptible the methods are to errors and LD heterogeneity. Methods based on signed LD correlations are more prone to errors and LD heterogeneity than those based on unsigned LD correlations or LD scores. For instance, COJO is relatively susceptible to errors and LD het-erogeneity because it relies on signed LD correlations in a mul-tiple regression framework. Suppose the effect allele of an SNP is mislabeled (equivalent to the effect size changed to the opposite direction) in the summary data. In that case, the effect sizes of the SNP and another SNP in LD with it will be heavily overestimated in a joint or conditional analysis because the input effect sizes and

LD pattern appear to suggest that the effects of the two SNPs mask each other in the marginal analysis. Even in the absence of errors in the summary data, the joint or conditional effects of two SNPs in LD can be estimated with substantial biases in the presence of large LD heterogeneity between the discovery and the reference. Compared to common variants, rare variants are less well imputed and thus are more likely to have LD heterogeneity between data sets, which may explain the remarkable improvement of COJO for rare variants before and after DENTIST QC.

Given that a QC step can potentially remove true signals, we make sure that DENTIST is conservative in filtering variants. We show by simulation that in the absence of errors and data heterogeneity, the DENTIST test statistics were not inflated, and on average, only <0.05% variants were filtered out by DENTIST (Supplementary Fig. 1). In practice, we implemented an iterative variant partitioning approach to avoid widespread inflation of the DENTIST statistics in the presence of data errors or heterogeneity. Throughout all the analyses performed in this study, we found in no cases DENTIST degraded the results, and DENTIST often only needed to remove a very small proportion of the variants to alleviate the biases (Supplementary Tables 3, 9, 10, 12).

Our method is an early attempt of QC for summary-data-based analyses. To avoid misuse, we summarize the usages and limitations, in addition to the features mentioned above. First, DENTIST is a QC method for detecting not only errors in summary data but also heterogeneity between discovery and reference data. As shown in our simulation, DENTIST does not guarantee to filter all the errors but most of them with large GWAS z-scores and a large proportion of them with small z-scores. Second, DENTIST can identify errors that passed the standard QC approaches (such as HWE test and allelic frequency checking), which makes it a complementary method to existing QC filters. We suggest that DENTIST-based QC should be applied after the standard QC as DENTIST is more powerful when the proportion of errors is smaller (Supplementary Table 2). Third, prior work suggests that the reference sample size ($n_{ref}$) needs to scale with GWAS sample size[49]. We have quantified the performance of DENTIST with respect to $n_{ref}$ and shown that an $n_{ref}$ of >5000 is recommended and the relationship between the performance of DENTIST and $n_{ref}$ does not seem to depend on GWAS sample size (Supplementary Fig. 2). Fourth, we may use DENTIST to choose from multiple LD reference samples the best matching reference for an analysis that requires reference LD. It can also be used as a diagnostic tool for sanity check of the GWAS summary data from a meta-analysis to identify potential issues in the contributing data sets or analysis pipeline. Lastly, DENTIST assumes the test statistics of different variants have similar sample sizes; violation of this assumption will lead to variants with significantly different sample sizes being recognized as problematic variants by DENTIST.

In summary, we have proposed a QC approach to improve summary-data-based analyses that are potentially affected by errors in GWAS summary or LD reference data or heterogeneity between data sets. This method has been implemented in a user-friendly software tool DENTIST. The software tool is multi-threaded so that it is computationally efficient when enough computing resources are available. For example, when running each chromosome in parallel, it took <1 h to run DENTIST with 4 CPUs for all variants with MAF > 1% and <5 h for all variants with MAF > 0.01% (Supplementary Table 14).

## Methods

**The DENTIST test-statistic**. Given an ancestrally homogeneous sample of $n$ unrelated individuals genotyped or imputed at $m$ variants, an association analysis is carried out at each variant by performing a linear regression between the variant and a phenotype of interest. This provides a set of summary data that include the

estimate of variant effect, the corresponding standard error, and thereby the z-statistic. Under the null hypothesis of no association, the z-scores of $m$ variants follow a multivariate normal distribution, $\mathbf{Z} \sim N(\mathbf{0}, \boldsymbol{\Sigma})$ with $\mathbf{Z} = (Z_1, Z_2, \ldots, Z_m)'$, with $\boldsymbol{\Sigma}$ being a LD correlation matrix of the variants.

The aim of the proposed method is to test the difference between the observed z-statistic of a variant and the predicted value from adjacent variants. To achieve this goal, we use a sliding window approach to divide genome into 2 Mb segments with a 500 kb overlap between one another and randomly partition the variants in a segment into two subsets, S1 and S2, with similar numbers of variants; we then use variants in S2 to predict those in S1 and vice versa. According to previous studies[12,13], under the null hypothesis of no association, the distribution of z-statistic of a variant $i$ from S1, conditional on the observed z-scores of a set of variants from S2 is

$$Z_i | \mathbf{Z_t} = \mathbf{z_t} \sim N\left(\boldsymbol{\Sigma}_{it} \boldsymbol{\Sigma}_{tt}^{-1} \mathbf{z_t}, \Sigma_{ii} - \boldsymbol{\Sigma}_{it} \boldsymbol{\Sigma}_{tt}^{-1} \boldsymbol{\Sigma}_{it}'\right), \quad (2)$$

where $\boldsymbol{\Sigma}_{it}$ is a row vector denoting the correlation of z-scores between variant $i$ from S1 and variants $\mathbf{t}$ from S2, and $\boldsymbol{\Sigma}_{tt}$ is the correlation matrix of variants $\mathbf{t}$. We use the correlation matrix calculated from an ancestry-matched reference sample (denoted by $\mathbf{R}$) to replace that in the discovery sample if individual-level genotypes of in the discovery GWAS are unavailable. In this case, Eq. 2 can be rewritten as

$$Z_i | \mathbf{Z_t} = \mathbf{z_t} \sim N\left(\mathbf{R}_{it} \mathbf{R}_{tt}^{-1} \mathbf{z_t}, \mathbf{I} - \mathbf{R}_{it} \mathbf{R}_{tt}^{-1} \mathbf{R}_{it}'\right). \quad (3)$$

As shown in previous studies[12,13], $E(Z_i | \mathbf{z_t})$ can be used to predict $Z_i$, i.e., $\widetilde{z}_i = \mathbf{R}_{it} \mathbf{R}_{tt}^{-1} \mathbf{z_t}$. In a special case when there is only one variant in vector $\mathbf{t}$, this model is the same as the conditional model in CAVIAR[11]. Hence, we can use the statistic below to test the difference between the observed and predicted z-scores

$$T_{d(i)} = (z_i - \mathbf{R}_{it} \mathbf{R}_{tt}^{-1} \mathbf{z_t})^2 / (1 - \mathbf{R}_{it} \mathbf{R}_{tt}^{-1} \mathbf{R}_{it}') \quad (4)$$

which approximately follows a $\chi^2$ distribution with 1 degree of freedom. A deviation of $T_{d(i)}$ from $\chi_1^2$ can be attributed to (1) errors in the summary data; (2) errors in the reference data; or (3) heterogeneity between the two data sets. These are likely to be the major but not the only factors that cause a discrepancy between the observed and imputed z-statistics, and other factors such as covariate-adjustment may also lead to a difference although such a difference is likely to be negligible (Supplementary Fig. 13).

Using Eq. 4, the test statistic $T_d$ can be calculated for each variant in S1 given z-scores from S2. As in previous studies[12,13], the method is derived under the null hypothesis without imposing any specific assumption about the genetic architecture of the trait, although our simulations show that it performs well under the alternative hypothesis (Supplementary Fig. 1). The exact model (not practically usable) for predicting the z-statistic under the alternative and the difference between the exact and approximate models can be found in the Supplementary Note. We show by numerical calculation that under the alternative, the test-statistics computed based on the approximate model were highly consistent with those based on the exact model (Supplementary Fig. 14). Note that this approximation is distinct from that frequently made in summary data-based analyses, where LD data from a reference sample are used to approximate those in the discovery sample. Furthermore, we assessed DENTIST under a host of MAF- and LD-dependent architectures, ranging from oligogenic ($m_{causal} = 50$) to highly polygenic models ($m_{causal} = 15,000$, 13.1% of all the variants) and found that the DENTIST test was still well calibrated (Supplementary Fig. 15).

To improve computational stability, we do not include variants in near-perfect LD with variant $i$ (e.g., $r^2 > 0.95$) in the variant set to compute $\widetilde{z}_i$. However, even by doing so, the LD correlation matrix $\mathbf{R}_{tt}$ can be rank-deficient. Hence, we use a truncated singular value decomposition (SVD) approach[15] to compute a pseudoinverse of $\mathbf{R}_{tt}$, i.e., performing an eigen decomposition of $\mathbf{R}_{tt}$ and retaining only the first $k$ eigenvalues ranked by eigenvalues.

$$\mathbf{R}_{it} \mathbf{R}_{tt}^{-1} \mathbf{z_t} = \mathbf{R}_{it} \mathbf{R}_{tt}^{+} \mathbf{z_t} = \sum_{j=1,\ldots,k} (1/w_j)(\mathbf{R}_{it} \boldsymbol{\nu}_j)(\boldsymbol{\nu}_j' \mathbf{z_t}) \quad (5)$$

$$\mathbf{R}_{it} \mathbf{R}_{tt}^{-1} \mathbf{R}_{it}' = \mathbf{R}_{it} \mathbf{R}_{tt}^{+} \mathbf{R}_{it}' = \sum_{j=1,\ldots,k} (1/w_j)(\mathbf{R}_{it} \boldsymbol{\nu}_j)^2 \quad (6)$$

$\mathbf{R}_{tt}^{+}$ denotes the pseudo inversion of $\mathbf{R}_{tt}$. The scalars $w_1, \ldots, w_k$ correspond to the largest $k$ eigenvalues, and vectors $\boldsymbol{\nu}_1, \ldots, \boldsymbol{\nu}_k$ are the corresponding $k$ eigenvectors. Given $q = \text{rank}(\mathbf{R}_{tt})$, the suggested value of $k$ is $k \ll q$. Let $\theta_k = k/q$. We show by simulation that $\theta_k = 0.5$ appears to be a good choice, meanwhile a large reference sample size (e.g., $n_{ref} \geq 5000$) is need (Supplementary Fig. 2).

**The iterative variant partitioning approach**. A challenge in applying Eq. 4 to real data is that errors in $\mathbf{z_t}$ or discrepancy between $\mathbf{R}_{it}$ and $\boldsymbol{\Sigma}_{it}$ can affect the accuracy of predicting $\widetilde{z}_i$. We propose an iterative variant partitioning approach to mitigate this issue. That is, in each iteration, we randomly partition the variants into two sets, S1 and S2, predict the z-statistic of each variant in S1 using its adjacent variants in S2 and vice versa, and compute a $T_d$ statistic for each of the variants. To mitigate the problem that variants with higher GWAS test statistics tend to have higher DENTIST test statistics, we stratify all the variants into two groups by a GWAS cutoff $P$ value of 0.01 and rank the variants by $P_{GWAS}$ from the smallest to the largest in either group. We first apply a $P_{DENTIST}$ threshold of $5 \times 10^{-8}$ to variants with $P_{GWAS} > 0.01$ and compute the proportion of variants removed (denoted by x

%). We apply the same $P_{DENTIST}$ threshold to variants with $P_{GWAS} < 0.01$ but capped the proportion of variants removed at the top x%. We impose an additional constraint that the proportion of variants removed is capped at the top 0.5% in either group. The default number of iterations is set to 10. In this iterative process, variants with large errors or LD heterogeneity between the discovery and LD reference samples are prioritized for removal in the first few iterations so that the prediction accuracy increases in the following iterations. After the iterations are completed, we remove all variants with $P_{DENTIST} < 5 \times 10^{-8}$ in the final step without any of the constraints.

**Genotype data sets**. This study is approved by the University of Queensland Human Research Ethics Committee (approval number: 2011001173) and the Westlake University Ethics Committee (approval number: 20200722YJ001). A summary of the genotype data sets used in this study as well as their relevant information can be found in Table 1. These data are from four GWAS cohorts of European descendants, including the Health Retirement Study (HRS)[35], Athero-sclerosis Risk in Communities (ARIC) study[36], UK10K[29], and UK Biobank (UKB)[31]. The samples were genotyped using either WGS or SNP array technology (Table 1). Imputation of the UKB data had been performed in a previous study[50] using the Haplotype Reference Consortium (HRC)[37] and UK10K reference panels[31,51]. We used different subsets of the imputed UKB data as the LD reference in this study, denoted with the prefix "UKBv3", such as UKBv3-unrel (all the unrelated individuals of European ancestry, $n = 348,577$), UKBv3-329K (a subset of 328,577 individuals of UKBv3-unrel), UKBv3-20K (another subset of 20,000 individuals of UKBv3-unrel, independent of UKBv3-392K) and UKBv3-8K (a subset of 8,000 individuals of UKBv3-20K). HRS, ARIC and UK10K cohorts were imputed to the 1KGP reference panel in prior studies[30,33], and a subset of 8,000 unrelated individuals from UKB were imputed to the 1KGP reference panel in this study (referred to as UKB-8K-1KGP). The UK10K variants in common with those on an Illumina CoreExome array were used for 1KGP imputation[33]. The imputation dosage values were converted to best-guess genotypes in all the data sets except for UKBv3-all, in which the hard-called genotypes were converted from the imputation dosage values using PLINK v1.90b3.38—hard-call-threshold 0.1 (ref. [52]). For all the data sets, standard QC was performed to remove variants with HWE test $P$ value $< 10^{-6}$, imputation INFO score $<0.3$, or MAF $< 0.001$. Since the hard-called genotypes had missing values, in UKBv3 and its subsets, we further removed variants with missingness rate $> 0.05$.

**Genome-wide association analysis for height using the UKB data**. We performed a genome-wide association analysis for height using the genotype data of UKBv3-329K, i.e., all the unrelated individuals of European ancestry in the UKB ($n = 328,577$) except for 20,000 individuals randomly selected to create a non-overlapping reference sample (i.e., UKBv3-20K). The height phenotype was pre-adjusted for sex and age. We conducted the association analysis using the simple linear regression model in the fastGWA[53] module of GCTA v1.92.3 with the first 10 principle components (PCs) fitted as covariates.

**Reporting summary**. Further information on research design is available in the Nature Research Reporting Summary linked to this article.

## Data availability

All the data sets used in this study are available in the public domain. The UKB data are available through the UK Biobank Access Management System (https://www.ukbiobank.ac.uk/). The HRS and ARIC data are available in the dbGaP database under accession numbers phs000428 and phs000280, respectively. The UK10K data are available in the EGA database under accession numbers EGAS00001000108 and EGAS00001000090. The 1KGP data are available at https://www.internationalgenome.org. The GWAS summary statistics are available with unrestricted access through the links provided in the corresponding publications listed in Table 4. Source data are provided with this paper.

## Code availability

The software tool DENTIST was written in C++ as a command-line tool. The source code and pre-compiled executable for 64-bit Linux distributions are available at https://doi.org/10.5281/zenodo.5516202 (ref. [54]).

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

## Acknowledgements

We are very grateful for constructive comments from Naomi Wray, Loic Yengo, Ying Wang and Jian Zeng and technical supports from Allan McRae, Julia Sidorenko, and Futao Zhang. This research was supported by the Australian Research Council (FT180100186, FL180100072), the Australian National Health and Medical Research Council (1113400, 1107258), the Sylvia & Charles Viertel Charitable Foundation, and the Westlake Education Foundation. This study makes use of data from the UK Biobank (application: 12505), HRS, ARIC and UK10K. A full list of acknowledgements to these data sets can be found in the Supplementary Note 2.

## Author contributions

J.Y. conceived and supervised the study. W.C., Z.Z.h. and J.Y. developed the method. W.C., Y.W., Z.Z.h. and J.Y. designed the experiment. W.C. performed the simulations and data analyses under the assistance and guidance from Y.W., Z.Z.l., T.Q., P.M.V., Z.Z.h. and J.Y. W.C. developed the software tool. P.M.V. and J.Y. contributed funding and resources. W.C. and J.Y. wrote the manuscript with the participation of all authors. All authors reviewed and approved the final manuscript.

## Competing interests

The authors declare no competing interests.

## Additional information

**Peer Review Information** *Nature Communications* thanks Farhad Hormozdiari, Martin Zhang, and the other, anonymous, reviewer(s) for their contribution to the peer review of this work. Peer reviewer reports are available.

