## [Peer Review File · Nature Communications]

Reviewer comments, response- -

Reviewer #1 (Remarks to the Author):

Many methods use both GWAS summary statistics and a reference LD panel. However, those methods may be biased by errors in the GWAS summary data and heterogeneity between the GWAS and the LD reference populations. This paper proposed a new quality control (QC) approach, namely DENTIST, that detects SNPs where such bias exists and filters those SNPs out. Through simulations, the authors showed that DENTIST can substantially reduce the false-positive rate in COJO association analysis. The authors also showed that DENTIST can improve other summary-data-based analyses such as LD score regression analysis, and integrative analysis of GWAS and expression quantitative trait locus data. Particularly, the method is more effective for rare variants, where the difference between the GWAS population and the LD reference population is larger.

Overall, the paper addresses an important problem in summary-statistics-based methods, namely the discrepancy between the in-sample LD and the LD estimated from a reference panel. The method seems to be effective, though filtering out problematic SNPs is a less fundamental solution as compared to improving the LD estimates from the reference panel. In addition, the empirical results are also extensive and convincing. Please see my detailed comments below.

Major

- According to lines 432-433, the authors seem to assume a single-component random-effect model where the SNP effects are i.i.d. with the same per-SNP variance. First, it is important to be explicit about this assumption as it will inform users when the null distribution of the test statistic $T_d(i)$ is valid and more generally when this method is appropriate to use. Second, if the authors have indeed assumed a single-component random-effect model, does it mean that the method only works for polygenic traits but not Mendelian traits? Some discussions on this would be helpful. Third, since the SNP effects of a real polygenic trait depend on both MAF and LD, it is important to investigate how the method would behave under a MAF-and-LD dependent genetic architecture. Some simulations and discussions on this would be helpful.

- The authors argued that the method does not hurt the power by showing that the method does not remove a lot of SNPs (<1%) in all cases. However, it may be possible that the method has removed a large proportion of causal SNPs. To eliminate this possibility, perhaps the authors can also supplement the percentage of high Z-score SNPs removed by the method.

- Line 438, the authors used an SNP-splitting strategy that randomly partitions the SNPs in a segment into two subsets. Such a procedure will introduce some randomness to the method. Since it is desirable for a QC method to be deterministic, it is important to quantify the consistency between different runs of the method. Please note that fixing a random seed would not help here because if I remove a handful of SNPs, I would still expect the result to be mostly the same. Furthermore, will a leave-one-out partitioning strategy, i.e., estimating the Z-score of one SNP using all other SNPs in the segment, be better? First, it is a deterministic approach. Second, most SNPs are used for prediction and hence such a procedure would have a higher prediction accuracy. Third, it will not be much slower because the intermediate quantities used for predicting different SNP Z-scores are similar and can be reused.

- The bias described in the paper seems to depend on the reference LD panel, the SNPs considered, and the downstream analysis. For example, the improvement is large for rare variants but small for common variants. Also, the improvement in the COJO example seems to be large, but those in the HEIDI example and the LDSC example are less so. It would be helpful to discuss when the DENTIST QC filtering is most useful.

- It seems that there is a mismatch between the text and Figure 3b. Specifically, in line 282, the author mentioned that the FPR is 5.8% for the one-sample scenario. However, the corresponding

Figure 3b seems to be showing something else.

Minor

- Typo in line 274: "a one-sample"  "an one-sample"
- Typo in line 409: "Four"  "Fourth"

Signed

Martin Jinye Zhang, PhD
Postdoctoral Researcher
Department of Epidemiology
Harvard T.H. Chan School of Public Health

Reviewer #2 (Remarks to the Author):

Chen et al. propose a QC procedure, DENTIST, to facilitate the summary data-based association analysis methods. Although I fully appreciate the authors' attention to this important practical issue, I find several technical aspects in the current manuscript are concerning. I hope that my comments help the authors revise the paper.

Major comments

1. The model assumptions and the interpretation of "error" need to be clarified. The idea of imputing z-scores using a set of anchor SNPs based on equations (1) and (2) is not new. It has been widely used in software packages like impG-summary (Pasaniuc et al. 2014). The claimed novelty in this paper is to inspect the discrepancy between the observed and imputed z-scores. However, I take an issue to interpret the discrepancy as an error. As the authors rightly noted on line 431 of the Methods section, the model is based on the null hypothesis of no association. Essentially, the Eqns (1) and (2) become incorrect if the target SNP i is indeed associated with the trait of interest. Noticeably, a noncentrality parameter is missing from the conditional mean. As a result, if a SNP is genuinely associated, the imputed z-score is expected to be different. (Mathematically, this is easy to see if Σ_{it} is $\rightarrow 0$ element-wise, and $S1$ contains a single genuinely associated SNP.) Hence, I would worry about interpreting the observed discrepancy as an error. At the minimum, there should be discussions on why and how the QC procedure will not remove the genuine association signals in such scenarios. (I fully understand that the authors did not find this as an issue in their simulations. But one could create another set of simulations to highlight this caveat. Thus, theoretical discussions are more relevant to address the issue at hand, IMO). As a reference, I suggest that the authors discuss the statistical model used in CAVIAR, where the noncentrality parameters under the alternative are explicitly modeled.
2. I am not sure that LD is the *only* factor that causes a discrepancy between observed and imputed Z-scores. Particularly, the observed Z-scores may be obtained by controlling some fixed effects in genetic association analysis, e.g., gender, age, genotype PCs. I don't see how these factors are accounted for in the imputation. Would these factors also explain some of the discrepancies? If so, could pursue concordance by solely focusing on LD yield misleading results?
3. I don't understand the claim that truncating the eigenvalues of R_{tt} is equivalent to "accounting for sampling noise in LD". Yes, the operation improves the computational stability, but what is the sampling noise the authors refer to? More specifically, what is the "true" population? The panel or the sample? Why modifying the panel correlation improves its match to the study sample?? Additionally, the LD pruning procedure described in Line 489 to 493 is vague and confusing. The whole paragraph needs to be better explained.
4. In the experiment with simulated phenotypes, the authors describe that the FPR ($> 28\%$) is greatly inflated for rare-causal variants. After applying the DENTIST, the corresponding FPR is

reduced to $< 2\%$. As the authors noted in the same section (line 194-195), the imputations are generally poor for rare-causal variants. I wonder if the dramatic reduction of FPR for this class of variants serves evidence for the proposed method's success. For comparison, if a QC procedure simply removes all the significant signals from rare variants, would the FPR be comparable? (I also note that this very reduction statistic is highlighted in the abstract but with incomplete descriptions on the variant class).

5. The HEIDI test simulation is problematic. The authors simulate a set of pleiotropic models with causal eQTLs and causal GWAS hits completely overlapping. They also generate a set of linkage models where causal eQTLs and causal GWAS hits are in LD ($r^2 > 0.25$). I wonder if r^2 is $\rightarrow 1$, how could the pleiotropic model and the linkage model be distinguished?? Do the authors mean they set an upper bound for r^2 in the linkage model? In that case, I find the simulation is overly simplified and lacks practical merit. As a result, I find the FPR and power numbers hard to understand.

Minor comments

Some of the notations are used without introduction. e.g., P_{dentist} , q^2 , etc.

Reviewer #3 (Remarks to the Author):

The authors have proposed a statistical method, DENTIST, to remove variants from GWAS in which their summary statistics and provided LD does not follow the expected value. Although the work is interesting, there are a couple of major concerns:

1) The Equation 1 in the paper, which is based on Imp-G and DIST, assumes most loci are under the NULL. However, due to high polygenicity and/or omnigenic model we do not believe this is true anymore. Wu et al. *Journal of Computational Biology* 2019 have shown how to extend the model to cases where a locus can have one or more causal variants. The authors need to explore this direction or at least discuss in the discussion. It is worth noting that applying DENTIST is more appealing to a locus with causal variants for downstream analysis than a null locus with no causal variant.

2) Regarding the point (1), I want to see an experiment where in multiple loci, the authors have implanted one, two, and three causal variants and DENTIST is applied to each locus to quantify the fraction of times DENTIST removes the true causal variants.

3) I like to see similar experiments to Figure 2 in which COJO FPR is compared where the authors implant more than one causal variant in the locus.

4) I would love to see the effect of DENTIST when applied on top of fine-mapping methods. I would recommend the authors to simulate loci with one, two and more causal variants and compare fine-mapping methods (FINEMAP or SuSiE) with and without DENTIST.

5) According to the authors' results for S-LDSC the h^2_g does not change and the change of intercept is not significant based on the numbers provided by authors (1.24 (se=0.04) vs 1.13 (se=0.04)). Thus, applying DENTIST to S-LDSC does not improve the results significantly. I recommend that the authors change the subsection heading or show results with a significant improvement.

Minor points:

1) Benner et al. *AJHG* 2017 have shown that in the context of fine-mapping using a reference panel with the same order of individuals is vital to avoid misleading results. The authors need to cite and elaborate in discussion about this paper. Particularly, most of the examples in the paper are based on sample size different between original GWAS LD and the LD panel used for downstream analysis.

2) Please use commas for numbers to be consistent throughout the paper. Either use a comma or do not use it.

3) The following papers need to be cited in the context of summary statistics:

Benner et al. AJHG 2017
Hormozdiari et al. Genetics 2014
Wang et al. Journal of the Royal Statistical Society 2020

RESPONSES TO REVIEWERS' COMMENTS

Re: We thank the reviewers for their constructive comments which have helped to improve our manuscript. We have responded to all the comments point by point below (in blue) in this document and highlighted all the relevant changes in yellow in the revised manuscript files. Here is a summary of the main changes.

- 1) As per the suggestion from reviewer #2, we have applied DENTIST to FINEMAP and shown a substantial improvement for FINEMAP after DENTIST QC when using out-of-sample LD.
- 2) We have demonstrated by additional analyses that the performance of DENTIST is robust to the randomness of variant splitting in the iterative process and whether the GWAS summary statistics have been adjusted for SNP-derived principal components in the presence of population stratification effect.
- 3) We observed a slight trend that variants with larger GWAS test statistics are more likely to be removed by DENTIST. To mitigate this issue, we have added a step to the iterative variant partitioning approach to harmonize the proportion of variants removed in two variant groups stratified by GWAS p-values in the initial iterations. The proportion of causal variants filtered has now been well controlled below 1%. We have also re-run all the analyses by this new DENTIST version, and the results remained largely unchanged.

Reviewer #1 (Remarks to the Author):

Many methods use both GWAS summary statistics and a reference LD panel. However, those methods may be biased by errors in the GWAS summary data and heterogeneity between the GWAS and the LD reference populations. This paper proposed a new quality control (QC) approach, namely DENTIST, that detects SNPs where such bias exists and filters those SNPs out. Through simulations, the authors showed that DENTIST can substantially reduce the false-positive rate in COJO association analysis. The authors also showed that DENTIST can improve other summary-data-based analyses such as LD score regression analysis, and integrative analysis of GWAS and expression quantitative trait locus data. Particularly, the method is more effective for rare variants, where the difference between the GWAS population and the LD reference population is larger.

Overall, the paper addresses an important problem in summary-statistics-based methods, namely the discrepancy between the in-sample LD and the LD estimated from a reference panel. The method seems to be effective, though filtering out problematic SNPs is a less fundamental solution as compared to improving the LD estimates from the reference panel. In addition, the empirical results are also extensive and convincing. Please see my detailed comments below.

Re: We thank the reviewer for the summary of our study and the positive remarks.

- According to lines 432-433, the authors seem to assume a single-component random-effect model where the SNP effects are i.i.d. with the same per-SNP variance. First, it is important to be explicit about this assumption as it will inform users when the null distribution of the test statistic $T_d(i)$ is

valid and more generally when this method is appropriate to use. Second, if the authors have indeed assumed a single-component random-effect model, does it mean that the method only works for polygenic traits but not Mendelian traits? Some discussions on this would be helpful. Third, since the SNP effects of a real polygenic trait depend on both MAF and LD, it is important to investigate how the method would behave under a MAF-and-LD dependent genetic architecture. Some simulations and discussions on this would be helpful.

Re: As stated in the Methods section (page 14) “Under the null hypothesis of no association, the z-scores of m variants follow ...”, our method is derived under the null hypothesis of no association, meaning that we do not make any specific assumption about the genetic architecture of the trait (we have made this clearer in the revised manuscript, page 16), and our simulations show that the method works well under the alternative hypothesis (Supplementary Figure 2). Nevertheless, as per the reviewer’s suggestion, we have now tested the performance of the method under a host of MAF- and LD-dependent architectures, ranging from oligogenic to highly polygenic models.

We have revised manuscript as follows (page 15).

“As in previous studies^{11,12}, the method is derived under the null hypothesis without making any specific assumption about the genetic architecture of the trait, although our simulations show that the method works well under the alternative hypothesis (**Supplementary Figure 1**). We show in the **Supplementary Note** the exact model (not practically usable) for predicting the z-statistic under the alternative, and the difference between the exact and approximate models. Our simulation shows that under the alternative, the test-statistics computed based on the approximate model were highly consistent with those based on the exact model (**Supplementary Figure 14**). Furthermore, we assessed DENTIST under a host of MAF- and LD-dependent architectures, ranging from oligogenic ($m_{\text{causal}} = 50$) to highly polygenic models ($m_{\text{causal}} = 15,000$, 13.1% of all the variants) and found that the DENTIST test statistic was still well calibrated (**Supplementary Figure 15**).”

- The authors argued that the method does not hurt the power by showing that the method does not remove a lot of SNPs (<1%) in all cases. However, it may be possible that the method has removed a large proportion of causal SNPs. To eliminate this possibility, perhaps the authors can also supplement the percentage of high Z-score SNPs removed by the method.

Re: We thank the reviewer for this comment. We did observe a slight trend that variants with higher GWAS test statistics were more likely to be removed, possibly because variants with higher GWAS test statistics are more prone to LD heterogeneity, leading to a slightly elevated proportion of causal variants removed (2.6% for causal variants vs. <1% for all variants). We have adjusted the iterative variant partitioning approach in DENTIST to mitigate this problem. That is, we stratify all the variants into two groups by a GWAS cutoff p-value of 0.01 and rank the variants by P_{GWAS} from the smallest to the largest in either group. We first apply a P_{DENTIST} threshold of 5×10^{-8} to variants with $P_{\text{GWAS}} > 0.01$ and compute the proportion of variants removed (denoted by $x\%$). We apply the same P_{DENTIST} threshold to variants with $P_{\text{GWAS}} < 0.01$ but capped the proportion of variants removed

at the top x%. We impose an additional constraint that the proportion of variants removed is capped at the top 0.5% in either group.

We repeat this variant stratification strategy until the final iteration where we remove all variants with $P_{\text{DENTIST}} < 5 \times 10^{-8}$ without the constraints. We show by simulation that the proportion of causal variants removed has now been well controlled below 1%. We have also re-run all the analyses with this new DENTIST version. All the results (including the power of detecting allelic and genotyping errors and the false-positive rate of COJO) remained largely unchanged.

We have made the changes below in the main text (pages 15-16).

“To mitigate the problem that variants with higher GWAS test statistics tend to have higher DENTIST test statistics, we stratify all the variants into two groups by a GWAS cutoff p-value of 0.01 and rank the variants by P_{GWAS} from the smallest to the largest in either group. We first apply a P_{DENTIST} threshold of 5×10^{-8} to variants with $P_{\text{GWAS}} > 0.01$ and compute the proportion of variants removed (denoted by x%). We apply the same P_{DENTIST} threshold to variants with $P_{\text{GWAS}} < 0.01$ but capped the proportion of variants removed at the top x%. We impose an additional constraint that the proportion of variants removed is capped at the top 0.5% in either group.”

“The percentage of causal variants removed from DENTIST was 0.19% using in-sample LD from UK10K-WGS, lower than 0.99% using out-of-sample LD from UKB-8K-1KGP (Supplementary Table 4).”

- Line 438, the authors used an SNP-splitting strategy that randomly partitions the SNPs in a segment into two subsets. Such a procedure will introduce some randomness to the method. Since it is desirable for a QC method to be deterministic, it is important to quantify the consistency between different runs of the method. Please note that fixing a random seed would not help here because if I remove a handful of SNPs, I would still expect the result to be mostly the same. Furthermore, will a leave-one-out partitioning strategy, i.e., estimating the Z-score of one SNP using all other SNPs in the segment, be better? First, it is a deterministic approach. Second, most SNPs are used for prediction and hence such a procedure would have a higher prediction accuracy. Third, it will not be much slower because the intermediate quantities used for predicting different SNP Z-scores are similar and can be reused.

Re: The nature of the method (i.e., predicting the test statistic of an SNP from other SNPs) determines that the output is not expected to be identical if the input SNPs are different, a common issue for all methods that model multiple SNPs jointly accounting for LD. The leave-one-out strategy would be a good choice in the absence of errors and LD heterogeneity. However, in the presence of errors and LD heterogeneity, the prediction of a target SNP can be affected by errors and LD heterogeneity in SNPs used for prediction. This is the reason why we developed the iterative variant partitioning approach to minimize errors and LD heterogeneity in the SNP set used for prediction/imputation.

Nevertheless, in the revised manuscript, we have quantified the robustness of the method to the randomness SNP splitting by using three different random seeds. The power of DENTIST to detect genotyping and allelic errors remained similar for different random seeds.

We have added the following statement in the revised manuscript (page 5).

“In addition, the power of DENTIST to detect genotyping and allelic errors was robust to randomness of variant splitting (**Supplementary Figure 3**), ...”

- The bias described in the paper seems to depend on the reference LD panel, the SNPs considered, and the downstream analysis. For example, the improvement is large for rare variants but small for common variants. Also, the improvement in the COJO example seems to be large, but those in the HEIDI example and the LDSC example are less so. It would be helpful to discuss when the DENTIST QC filtering is most useful.

Re: We thank the reviewer for the suggestion and have added the following statements in the Discussion section (page 12).

“In general, improvement of a summary data-based analysis before and after DENTIST QC depends on how susceptible the methods are to errors and LD heterogeneity. Methods based on signed LD correlations are more prone to errors and LD heterogeneity than those based on unsigned LD correlations or LD scores. For instance, COJO is relatively susceptible to errors and LD heterogeneity because it relies on signed LD correlations in a multiple regression framework. Suppose the effect allele of an SNP is mislabeled (equivalent to the effect size changed to the opposite direction) in the summary data. In that case, the effect sizes of the SNP and another SNP in LD with it will be heavily overestimated in a joint or conditional analysis because the input effect sizes and LD pattern appear to suggest that the effects of the two SNPs mask each other in the marginal analysis. Even in the absence of errors in the summary data, the joint or conditional effects of two SNPs in LD can be estimated with substantial biases in the presence of large LD heterogeneity between the discovery and the reference. Compared to common variants, rare variants are less well imputed and thus are more likely to have LD heterogeneity between data sets, which may explain the remarkable improvement of COJO for rare variants before and after DENTIST QC.”

- It seems that there is a mismatch between the text and Figure 3b. Specifically, in line 282, the author mentioned that the FPR is 5.8% for the one-sample scenario. However, the corresponding Figure 3b seems to be showing something else.

Re: We thank the reviewer for spotting this mistake. We have corrected it in the revised manuscript (see Figure 4).

- Typo in line 274: "a one-sample"  "an one-sample"

Re: Corrected.

- Typo in line 409: "Four"  "Fourth"

Re: Corrected.

Reviewer #2 (Remarks to the Author):

Chen et al. propose a QC procedure, DENTIST, to facilitate the summary data-based association analysis methods. Although I fully appreciate the authors' attention to this important practical issue, I find several technical aspects in the current manuscript are concerning. I hope that my comments help the authors revise the paper.

Major comments

1. The model assumptions and the interpretation of "error" need to be clarified. The idea of imputing z-scores using a set of anchor SNPs based on equations (1) and (2) is not new. It has been widely used in software packages like impG-summary (Pasaniuc et al. 2014). The claimed novelty in this paper is to inspect the discrepancy between the observed and imputed z-scores. However, I take an issue to interpret the discrepancy as an error. As the authors rightly noted on line 431 of the Methods section, the model is based on the null hypothesis of no association. Essentially, the Eqns (1) and (2) become incorrect if the target SNP i is indeed associated with the trait of interest. Noticeably, a noncentrality parameter is missing from the conditional mean. As a result, if a SNP is genuinely associated, the imputed z-score is expected to be different. (Mathematically, this is easy to see if Σ_{it} is $\rightarrow 0$ element-wise, and $S1$ contains a single genuinely associated SNP.) Hence, I would worry about interpreting the observed discrepancy as an error. At the minimum, there should be discussions on why and how the QC procedure will not remove the genuine association signals in such scenarios. (I fully understand that the authors did not find this as an issue in their simulations. But one could create another set of simulations to highlight this caveat. Thus, theoretical discussions are more relevant to address the issue at hand, IMO). As a reference, I suggest that the authors discuss the statistical model used in CAVIAR, where the noncentrality parameters under the alternative are explicitly modeled.

Re: The reviewer is correct that our method is an approximation, under an implicit assumption that the expected value of the z-statistic of a target SNP under the alternative hypothesis can be perfectly predicted by the expected values of the z-statistics of SNPs used for prediction. Note that this is also the assumption underlying ImpG-Summary (Pasaniuc et al. 2014 Bioinformatics). In the revised manuscript, we have now provided in the **Supplementary Note** the exact form of the conditional distribution and have shown explicitly the difference between the approximate and exact models. We have further shown by simulation that the DENTIST test-statistics based on the approximate model are highly consistent with those based on the exact model (**Supplementary Figure 14**). This result may also explain why ImpG-Summary works well for GWAS data under the alternative. In addition, as per the suggestion from Reviewer #1 (see above), we have reported the

proportion of causal variants removed by DENTIST in **Supplementary Figure 1** and **Supplementary Table 4**, showing that the proportion of causal variants removed by DENTIST was very small (<1%).

We have added the following statements in the main text (page 15).

“As in previous studies^{11,12}, the method is derived under the null hypothesis without imposing any specific assumption about the genetic architecture of the trait, although our simulations show that the method performs well under the alternative hypothesis (**Supplementary Figure 1**). The exact model (not practically usable) for predicting the z-statistic under the alternative, and the difference between the exact and approximate models can be found in the **Supplementary Note**. We show by numerical calculation that under the alternative, the test-statistics computed based on the approximate model were highly consistent with those based on the exact model (**Supplementary Figure 14**).”

The multivariate distribution of the test-statistics for a set of SNPs in LD has been shown in different forms in the literature. For example, in the CAVIAR paper (Hormozdiari, F. Genetics, 2014), the authors model the joint distribution of the test-statistics with an explicit expression of the non-centrality parameters. However, it is interesting to note that CAVIAR also uses the model under the null for prediction. In fact, the CAVIAR conditional model is a special case of our approximate method (i.e., the ImpG-Summary model) when there is only one SNP in vector \mathbf{t} .

We have included a statement in the revised manuscript to comment on this (page 14).

“In this case, **Equation 1** can be rewritten as

$$Z_i|\mathbf{z}_t \sim N(\mathbf{R}_{it}\mathbf{R}_{tt}^{-1}\mathbf{z}_t, \mathbf{1} - \mathbf{R}_{it}\mathbf{R}_{tt}^{-1}\mathbf{R}'_{it}) \quad (2).$$

As shown in previous studies^{11,12}, $E(Z_i|\mathbf{z}_t)$ can be used as a predictor for Z_i , i.e., $\tilde{z}_i = \mathbf{R}_{it}\mathbf{R}_{tt}^{-1}\mathbf{z}_t$. In a special case when there is only one variant in vector \mathbf{t} , this model is the same as the conditional model in CAVIAR⁵⁰. We can use $E(Z_i|\mathbf{z}_t)$ as a predictor of Z_i , i.e., $\tilde{z}_i = \mathbf{R}_{it}\mathbf{R}_{tt}^{-1}\mathbf{z}_t$, and can therefore use the test-statistic below to test the difference ...”

2. I am not sure that LD is the *only* factor that causes a discrepancy between observed and imputed Z-scores. Particularly, the observed Z-scores may be obtained by controlling some fixed effects in genetic association analysis, e.g., gender, age, genotype PCs. I don't see how these factors are accounted for in the imputation. Would these factors also explain some of the discrepancies? If so, could pursue concordance by solely focusing on LD yield misleading results?

Re: We do not claim that LD is the only factor that causes the discrepancy and have added a statement in the main text for clarification (page 14-15): “These are likely to be the major but not the only factors that causes a discrepancy between the observed and imputed z-statistics, and other factors such as covariate-adjustment may lead to a difference.”

We have assessed the effect of covariate adjustment on DENTIST test-statistic by simulation. We used the UK10K-WGS data to simulate a polygenic trait (10,000 causal variants explaining 20% of the phenotypic variance in total) with 5% of the variance explained by the first two SNP-derived principal components (PCs). (We did not include age and gender in the simulation because they are not expected to be associated with SNP genotypes in the absence of age and gender participation biases.) We found little difference between the DENTIST test-statistics with and without PC correction of the GWAS summary data (note that the reference sample is always not PC-corrected; **Supplementary Figure 4**), suggesting that difference in covariate-adjustment between discovery and reference has limited impact on DENTIST test.

We have added the following statement in the main text (page 5):

“In addition, the power of DENTIST to detect genotyping and allelic errors was robust to the randomness of variant splitting (**Supplementary Figure 3**), or whether the GWAS analysis was corrected for the variant-derived principal components in the presence of population stratification effect (**Supplementary Figure 4**).”

3. I don't understand the claim that truncating the eigenvalues of R_{tt} is equivalent to "accounting for sampling noise in LD". Yes, the operation improves the computational stability, but what is the sampling noise the authors refer to? More specifically, what is the "true" population? The panel or the sample? Why modifying the panel correlation improves its match to the study sample?? Additionally, the LD pruning procedure described in Line 489 to 493 is vague and confusing. The whole paragraph needs to be better explained.

Re: We thank the reviewer for pointing out this. The purpose of performing both the truncated SVD and LD pruning is to improve the computational stability. We have revised text (page 15) accordingly as follows.

“To improve computational stability, we do not include variants in near-perfect LD with variant i (e.g., $r^2 > 0.95$) in the variant set for predicting \tilde{z}_i . However, even by doing so, the LD correlation matrix \mathbf{R}_{tt} can be rank-deficient. Hence, we use a truncated singular value decomposition (SVD) approach¹⁴ to compute a pseudoinverse of \mathbf{R}_{tt} , i.e., performing an eigen decomposition of \mathbf{R}_{tt} and retaining only the first k eigenvalues ranked by eigenvalues.

$$\mathbf{R}_{it}\mathbf{R}_{tt}^{-1}\mathbf{z}_t = \mathbf{R}_{it}\mathbf{R}_{tt}^+\mathbf{z}_t = \sum_{1..k} 1/w_k(\mathbf{R}_{it}\mathbf{v}_k)(\mathbf{v}'_k\mathbf{z}_t) \quad (4)$$

$$\mathbf{R}_{it}\mathbf{R}_{tt}^{-1}\mathbf{R}'_{it} = \mathbf{R}_{it}\mathbf{R}_{tt}^+\mathbf{R}'_{it} = \sum_{1..k} 1/w_k(\mathbf{R}_{it}\mathbf{v}_k)^2 \quad (5)$$

\mathbf{R}_{tt}^+ denotes the pseudo inversion of \mathbf{R}_{tt} . The scalars $w_1...w_k$ correspond to the largest k eigenvalues, and vectors $\mathbf{v}_1...v_k$ are the corresponding k eigenvectors. Given $q = \text{rank}(\mathbf{R}_{tt})$, the suggested value of k is $k \ll q$. Let $\theta_k = k/q$. We show by simulation that $\theta_k = 0.5$ appears to be a good choice, meanwhile a large reference sample size (e.g., $n_{\text{ref}} \geq 5000$) is need (**Supplementary Figure 2**).”

4. In the experiment with simulated phenotypes, the authors describe that the FPR (> 28%) is greatly inflated for rare-causal variants. After applying the DENTIST, the corresponding FPR is

reduced to < 2%. As the authors noted in the same section (line 194-195), the imputations are generally poor for rare-causal variants. I wonder if the dramatic reduction of FPR for this class of variants serves evidence for the proposed method's success. For comparison, if a QC procedure simply removes all the significant signals from rare variants, would the FPR be comparable? (I also note that this very reduction statistic is highlighted in the abstract but with incomplete descriptions on the variant class).

Re: As stated in our manuscript (page 7) "... DENTIST filtering caused a <1% loss of power for common-causal and 3-4% for rare-causal (Table 1), which is acceptable because the reduction in FPR was larger than that in power, especially for rare variants.", the decrease in power (3-4%) for rare variants is smaller than the reduction of FPR (>24%), suggesting a relative gain. Moreover, we have demonstrated in the GWAS simulations that genotyping or allelic errors were highly enriched in the set of variants removed by DENTIST, with fold enrichment of 137 ± 17.0 and 366 ± 58.7 for the two types of errors respectively using out of sample LD. All the results suggest that DENTIST is certainly better than random guess and has made COJO applicable to rare variants (the FPR of COJO for rare variants without DENTIST was too high to be practically useful).

In the abstract, we have been explicit that the highlighted numbers are for rare variants: "..., especially for imputed rare variants (FPR reduced from >28% to <2% in the presence of heterogeneity between GWAS and LD reference)."

5. The HEIDI test simulation is problematic. The authors simulate a set of pleiotropic models with causal eQTLs and causal GWAS hits completely overlapping. They also generate a set of linkage models where causal eQTLs and causal GWAS hits in LD ($r^2 > 0.25$). I wonder if r^2 is $\rightarrow 1$, how could the pleiotropic model and the linkage model be distinguished?? Do the authors mean they set an upper bound for r^2 in the linkage model? In that is the case, I find the simulation is overly simplified and lacks practical merit. As a result, I find the FPR and power numbers hard to understand.

Re: The reviewer is correct that the power of HEIDI to distinguish between linkage and pleiotropic models is proportional to $1 - r^2$ and will eventually decrease to zero if r^2 approaches to one, an issue that has been well recognized and discussed in prior work (Zhu et al. 2016 Nat Genet; Wu et al. 2018 Nat Commun). In our simulations, we did not set an upper bound for r^2 in the linkage model and have clarified this in the revised manuscript (page 9).

"To simulate a linkage model, a second causal variant in LD with the causal variant ($0.25 < r^2 \leq 1$) was selected to generate the gene expression level, again with the eQTL q^2 value sampled from the CAGE."

Minor comments

Some of the notations are used without introduction. e.g., P_{dentist} , q^2 , etc.

Re: We have revised the relevant text as follows.

Main text (page 3): " T_d follows approximately a χ^2 distribution with 1 degree of freedom. We denote P -value calculated from T_d by P_{DENTIST} ."

Main text (page 4): "We simulated a trait affected by 50 common, causal variants with effects drawn from $N(0,1)$, which together explained 20% of the phenotypic variation, with the proportion of variance explained by each causal variant (denoted by q^2) of 0.4%, on average."

Reviewer #3 (Remarks to the Author):

The authors have proposed a statistical method, DENTIST, to remove variants from GWAS in which their summary statistics and provided LD does not follow the expected value. Although the work is interesting, there are couple of major concerns:

1) The Equation 1 in the paper, which is based on Imp-G and DIST, assumes most loci are under the NULL. However, due to high polygenicity and/or omnigenic model we do not believe this is true anymore. Wu et al. Journal of Computational Biology 2019 have shown how to extend the model to cases where a locus can have one or more causal variants. The authors need to explore this direction or at least discuss in the discussion. It is worth noting that applying DENTIST is more appealing to locus with causal variants for downstream analysis than Null locus with no causal variant.

Re: This comment relates to comment #1 from Reviewer #2. As in the response above, our method is an approximation, which is derived under the null, assuming that the expected value of the z-statistic of a target SNP can be perfectly imputed using the expected values of the z-statistics of SNPs used for prediction. We have now included in the Supplementary Note the exact model derived under the alternative and the difference between the exact and approximate models. We then show by simulation that the DENTIST test statistics computed using the approximate method are highly consistent with those computed using the exact method (Supplementary Figure 14).

We have added the following statements in the main text (page 15).

"The exact model (not practically usable) for predicting the z-statistic under the alternative, and the difference between the exact and approximate models can be found in the **Supplementary Note**. We show by numerical calculation that under the alternative, the test-statistics computed based on the approximate model were highly consistent with those based on the exact model (**Supplementary Figure 14**)."

We have further assessed the robustness of DENTIST by expanding the simulation to include more complicated patterns of genetic architecture (MAF- and LD-dependent architecture with a wide range of polygenicity, i.e., the proportion of variants being causal) and allelic heterogeneity (one,

two and three causal variants at a locus). The DENTIST test statistics remained well calibrated in these additional simulations (**Supplementary Figure 15**).

2) Regarding the point (1), I want to see an experiment where in multiple loci, the authors have implanted one, two, and three causal and DENTIST is applied to each locus to quantify the fraction of times DENTIST removes the true causal variants.

Re: As per the reviewer's suggestion, we have added in our simulations two additional scenarios with two and three quasi-independent causal variants ($r^2 < 0.1$ between the causal variants and $q^2 = 2\%$ for each causal variant), respectively, at a locus. We repeated the simulation 440 times. The fractions of simulation replicates in which DENTIST removed one or more causal variants were 0%, 1.6% and 1% for the scenarios with one, two and three causal variants, respectively (**Supplementary Figure 1**).

We have revised the relevant text as following (page 4).

"Prior to the simulations with errors, we performed a benchmark simulation (i.e., simulating a scenario without errors and applying DENTIST using the discovery GWAS sample as the LD reference) to show that the DENTIST test statistics were well calibrated, given the small fraction of variants removed by DENTIST in each simulation replicate and the small proportion of causal variants removed across simulation replicates, in the absence of errors and LD heterogeneity (**Supplementary Figure 1**)"

3) I like to see similar experiments to Figure 2 in which COJO FPR is compared where the authors implant more than one causal variant in the locus.

Re: The FPR in Figure 2 was quantified as the % of simulated loci identified with more than one COJO signal. In the revised manuscript, we have generalized this definition to be the % of simulated loci at which the number of COJO signals was larger than the number of simulated causal variants. Results from the simulation scenarios with two and three causal variants are presented in Supplementary Figure 7. The pattern of FPR changes before and after DENTIST QC in the two- and three-causal-variant scenarios was similar to that in the one-causal-variant scenario.

We have included the following sentence in the revised manuscript (page 6).

"We performed additional simulations with three causal variants at a locus and observed similar patterns of changes in FPR before and after DENTIST QC (**Supplementary Figure 7**)."

4) I would love to see the effect of DENTIST when applied on top of fine-mapping methods. I would recommend the authors to simulate loci with one, two and more causal variants and compare fine-mapping methods (FINEMAP or SuSiE) with and without DENTIST.

Re: We thank the reviewer for the suggestion and have added the following text to the revised manuscript.

“Improved FINEMAP analysis by DENTIST

FINEMAP⁹ is a method used for prioritizing causal variants from GWAS summary statistics. We assessed the effect of DENTIST QC on an FINEMAP analysis using the data simulated above with one, two or three common causal variants at a 1-Mb locus. The impact of the use of an out-of-sample LD reference on FINEMAP was evaluated by the estimated number of causal variants before or after DENTIST QC, in comparison with the benchmark analysis using in-sample LD. We observed that FINEMAP using out-of-sample LD tended to overestimate the number of causal variants (**Supplementary Figures 10 and 11**). For example, given a posterior inclusion probability threshold of 95%, FINEMAP with out-of-sample LD from the HRS cohort provided the correct number of causal variants in 78.4%, 49.8%, and 44.8% of the simulation replicates in the one-, two-, and three-causal scenarios, respectively (**Figure 3a**). After DENTIST QC, the proportions of times when FINEMAP identified the correct number of causal variants increased substantially to 87.3%, 78.5%, and 63.4% in the one-, two-, and three-causal scenarios, respectively, almost comparable to the corresponding proportions (i.e., 91.4%, 87.2%, and 68.2%) from the benchmark analysis (**Figure 3a**). The results were similar when using ARIC as the LD reference (**Figure 3b**). These results demonstrate a substantial improvement of the performance of FINEMAP after DENTIST QC.”

5) According to the authors’ results for S-LDSC the h^2g does not change and the change of intercept is not significant based on the numbers provided by authors (1.24 (se=0.04) vs 1.13 (se=0.04)). Thus, applying DENTIST to S-LDSC does not improve the results significantly. I recommend that the authors change the subsection heading or show results with a significant improvement.

Re: Done

We have revised the text as follows.

“LD score regression analysis with DENTIST”

Minor points:

1) Benner et al. AJHG 2017 have shown that in the context of fine-mapping using a reference panel with the same order of individuals is vital to avoid misleading results. The authors need to cite and elaborate in discussion about this paper. Particularly, most of the examples in the paper are based on sample size different between original GWAS LD and the LD panel used for down-stream analysis.

Re: Benner et al. (AJHG 2017) show that for a fine-mapping analysis that uses out-of-sample LD, the reference sample size (n_{ref}) needs to scale with GWAS sample size. We have quantified the performance of DENTIST with respect to n_{ref} and shown that a reference sample size of >5000 is recommended and the relationship between the performance of DENTIST and n_{ref} does not seem to depend on GWAS sample size (page 5).

As per the reviewer's suggestion, we have added a comment on this issue and cited the Benner et al. paper (page 13).

“Third, prior work suggests that n_{ref} needs to scale with GWAS sample size⁴⁹. We have quantified the performance of DENTIST with respect to reference sample size (n_{ref}) and shown that a reference sample size of >5000 is recommended and the relationship between the performance of DENTIST and n_{ref} does not seem to depend on GWAS sample size (**Supplementary Figure 2**).”

2) Please use comma for numbers to be consistent throughout the paper. Either use a comma or do not use it.

Re: Done.

3) The following papers need to be cited in the context of summary statistics:

Benner et al. AJHG 2017

Hormozdiari et al. Genetics 2014

Wang et al. Journal of the Royal Statistical Society 2020

Re: Done.

Reviewer comments, further- -

Reviewer #1 (Remarks to the Author):

I thank the authors for their effort and detailed response.

All my concerns have been satisfactorily addressed and I do not have further concerns.

Reviewer #2 (Remarks to the Author):

I appreciate the authors' efforts to address my previous comments. Here are some of my remaining concerns on the technical issues and claims in the paper.

1. As the authors explicitly acknowledged in their response that their proposed method is an **approximation**. But what is the method approximating? My understanding is that the approximation is to the analysis where full individual-level data are available. As a corollary, it implies that summary-level statistics-based information will not exceed the accuracy or the efficiency of individual-level data-based inference. For example, I would not understand the authors' generalizations, such as, "difference in covariate-adjustment between discovery and reference has limited impact on DENTIST test." Does it mean we don't need to control the covariates even in GWAS analysis when individual-level data are available? I also don't understand using summary-level GWAS information to detect genotyping and allelic errors, which are typically handled (quantified and reported) in the original individual-level data prior to any phenotype-related analysis. To be clear, I don't think the genotyping error is the major concern for summary-level statistics-based inference. The discrepancy from the analysis using individual-level data is the core issue here. I wish the authors could illustrate that their proposed approaches can effectively reduce such discrepancies.

2. The authors tend to address various comments by simulation evidence. I am naturally concerned about if the simulation settings used in different numerical experiments are generalizable. From this perspective, I find the authors' response to my previous comments (especially comments 1 and 2) less persuasive.

3. My concern about the HEIDI test is unaddressed. I was questioning the validity of the HEIDI test because the underlying models are not identifiable. I still don't understand why it illustrates the benefit of the new method. IMO, the experiment design is flawed.

Reviewer #3 (Remarks to the Author):

The authors have answered all my concerns. DENTIST can be a useful tool in the field.

RESPONSES TO REVIEWERS' COMMENTS

Re: We thank the reviewers again for their efforts reviewing our manuscript. We have responded to the additional comments from reviewer #2 point by point below and highlighted all the relevant changes in yellow in the manuscript files. For ease of reading, we have split comment #1 from reviewer #2 into two sub-comments.

Reviewer #1 (Remarks to the Author):

I thank the authors for their effort and detailed response. All my concerns have been satisfactorily addressed and I do not have further concerns.

Re: We thank the reviewer for the positive feedback.

Reviewer #2:

1.1 As the authors explicitly acknowledged in their response that their proposed method is an *approximation*. But what is the method approximating? My understanding is that the approximation is to the analysis where full individual-level data are available. As a corollary, it implies that summary-level statistics-based information will not exceed the accuracy or the efficiency of individual-level data-based inference. For example, I would not understand the authors' generalizations, such as, "difference in covariate-adjustment between discovery and reference has limited impact on DENTIST test." Does it mean we don't need to control the covariates even in GWAS analysis when individual-level data are available?

Re: First, it should be clarified that the primary aim of DENTIST is not to use GWAS summary statistics and reference LD to approximate any individual-level data-based analysis but to reduce heterogeneity between the GWAS and reference data to improve other summary data-based analyses (lines 342-345). As noted in our manuscript and previous response letter, we regard our method as an approximation because it is derived under the null hypothesis of no association. We have shown by theoretical derivation that the approximate and exact methods differ only by one term, i.e., $\mu_i - \Sigma_{it}\Sigma_{tt}^{-1}\mu_t$, where μ_i is the expected value of the GWAS test statistic for variant i , μ_t is a vector of the expected GWAS test statistics for an array of variants (t) used to predict variant i , Σ_{it} is a vector of LD correlations between variant i and the variants in t , and Σ_{tt} is the LD correlation matrix for the variants in t . Since $\Sigma_{it}\Sigma_{tt}^{-1}\mu_t$ can be regarded as a predictor of μ_i , the term $\mu_i - \Sigma_{it}\Sigma_{tt}^{-1}\mu_t$ is expected to be small even under the alternative, in line with our numerical analysis that under the alternative, the z-statistics predicted based on the approximate method are highly consistent with those predicted based on the exact method (**Supplementary Note**). We have revised the Supplementary Note accordingly for more clarification.

We agree with the reviewer that covariate adjustment could be another source of heterogeneity between the GWAS and LD reference samples, given that covariate adjustment is often performed in the GWAS but not in the LD reference. We reason that such a contribution is likely to be negligible, as suggested by the observation that LD correlations computed from the UK10K genotype data with covariate adjustment are almost identical to those without adjustment (**Supplementary Figure 13**). To recap that DENTIST is not a method to approximate any individual-level data-based analysis but a method to improve other summary data-based analyses by reducing heterogeneity between the discovery and reference samples. In this regard, even in the presence of

heterogeneity attributed to covariate adjustment, applying DENTIST to reduce such heterogeneity would still be beneficial for other summary data-based analyses.

We have revised the manuscript as follows (lines 345-350).

“There could be another source of heterogeneity if the GWAS analysis is adjusted for covariates such as age, sex, and genetic PCs, but the genotype data in the reference sample are not covariate-adjusted. However, such a contribution is likely to be negligible given the near perfect correlation between LD correlation values computed from the UK10K genotype data before and after covariate adjustment (**Supplementary Figure 13**).”

1.2 I also don't understand using summary-level GWAS information to detect genotyping and allelic errors, which are typically handled (quantified and reported) in the original individual-level data prior to any phenotype-related analysis. To be clear, I don't think the genotyping error is the major concern for summary-level statistics-based inference.

Re: We agree with the reviewer that genotyping errors are unlikely to be a major concern for summary data-based analyses. The purpose of adding genotyping errors to the simulation was to quantify the ability of DENTIST to detect errors in general and to explore the possibility of using DENTIST as an additional QC step for GWAS analyses with individual data to detect genotyping errors not detected by the standard QC pipeline. We have clarified this in the revised manuscript (lines 114-118).

2. The authors tend to address various comments by simulation evidence. I am naturally concerned about if the simulation settings used in different numerical experiments are generalizable. From this perspective, I find the authors' response to my previous comments (especially comments 1 and 2) less persuasive.

Re: Simulation is a widely used approach to calibrate a newly developed method because the grand truth is often unknown in real data. We appreciate the comment about generalizability and have made tremendous efforts to achieve this goal. First, all the simulations in this study were based on real genotype data so that real allele frequency spectrum and LD structure were used. For association analysis, we have accounted for the loss of information due to imputation by simulating causal variants on sequence variants but performing association tests using imputed genotype data. With respect to the genetic architecture of the simulated phenotypes, we have varied the number of causal variants from a relatively sparse genetic architecture to a highly polygenic architecture and have also considered multiple causal variants at a single locus. It is reassuring that the implications from real data analyses were by-and-large consistent with the simulation results. Hence, we believe that our simulation results are generalizable in general.

The reviewer's previous comment #1 was mainly about the derivation of the method. We responded that the method was derived under the null of no association, as in previous studies (including the one mentioned by the reviewer). We then showed, by theoretical derivation, the difference between the approximate and exact methods and confirmed by numerical calculation that the difference was small, consistent with expectation (Supplementary Note). These findings also explain why the summary data-based imputation methods derived under the null, such as ImpG-Summary (Pasaniuc et al. 2014 Bioinformatics), work reasonably well in practice.

The reviewer's previous comment #2 is highly related to comment #1.1 of this review. As we have clarified above, we agree that covariate adjustment could be another source of heterogeneity. However, the contribution is likely to be negligible given the almost

perfect correlation between LD values computed from the genotype data before and after covariate adjustment (**Supplementary Figure 13**). Note that even if there are differences in LD owing to covariate adjustment, applying DENTIST to eliminate some of the differences would still be beneficial for other summary data-based analyses.

3. My concern about the HEIDI test is unaddressed. I was questioning the validity of the HEIDI test because the underlying models are not identifiable. I still don't understand why it illustrates the benefit of the new method. IMO, the experiment design is flawed.

Re: As per the reviewer's comment, we have removed the whole part about the HEDI test from the manuscript.

Reviewer #3 (Remarks to the Author):

The authors have answered all my concerns. DENTIST can be a useful tool in the field.

Re: We thank the reviewer for the positive remark.

Reviewer comments, further response - -

Reviewer #1 (Remarks to the Author):

Overall I think the authors provided a reasonable response to reviewer 2's comments. Here are the details of my assessments.

1.1 I don't think I agree with the authors that "DENTIST does not approximate any individual-level data-based analysis". Specifically, I think it is fair to say that summary statistics-based methods, in general, try to approximate the individual-level data-based analyses. Therefore, DENTIST improves existing summary statistics-based methods and by doing so provides a better approximation of the individual-level data-based analyses.

I am also aware that the "approximation" discussed in the paper is a different kind, i.e., approximating the alternative distribution using the null distribution. It is helpful to clarify this in the corresponding places (that it is not about approximating the individual-level analyses).

1.2 I think the authors' response is appropriate.

2. I think it is widely accepted in the field to use extensive simulations to establish the validity of the method.

3. I think the authors' response is appropriate.

Signed

Martin Jinye Zhang, PhD
Postdoctoral Researcher
Department of Epidemiology
Harvard T.H. Chan School of Public Health

RESPONSES TO REVIEWERS' COMMENTS

Reviewer #1 (Remarks to the Author):

Overall I think the authors provided a reasonable response to reviewer 2's comments. Here are the details of my assessments.

1.1 I don't think I agree with the authors that "DENTIST does not approximate any individual-level data-based analysis". Specifically, I think it is fair to say that summary statistics-based methods, in general, try to approximate the individual-level data-based analyses. Therefore, DENTIST improves existing summary statistics-based methods and by doing so provides a better approximation of the individual-level data-based analyses.

Re: We agree and have removed this statement.

I am also aware that the "approximation" discussed in the paper is a different kind, i.e., approximating the alternative distribution using the null distribution. It is helpful to clarify this in the corresponding places (that it is not about approximating the individual-level analyses).

Re: The reviewer is correct. The approximation discussed in the paper refers to the use of the chi-squared test statistic derived under the null hypothesis of no association to approximate that under the alternative hypothesis. We have now clarified in the text that this approximation is distinct from that frequently made in summary data-based analyses, where LD data from a reference sample are used to approximate those in the discovery sample (lines 534-536).

1.2 I think the authors' response is appropriate.

2. I think it is widely accepted in the field to use extensive simulations to establish the validity of the method.

3. I think the authors' response is appropriate.

Re: We thank the reviewer for these remarks.